# Base editing of *Ptbp1* in neurons alleviates symptoms in a mouse model of Parkinson's disease

**Desiree Böck[1]\*[†], Maria Wilhelm[1][†], Jonas Mumenthaler[1], Daniel Fabio Carpanese[1], Peter I Kulcsár[1], Simon d'Aquin[1], Alessio Cremonesi[2], Anahita Rassi[2], Johannes Häberle[3], Tommaso Patriarchi[1,4]\*, Gerald Schwank[1]\***

[1]Institute of Pharmacology and Toxicology, University of Zurich, Zurich, Switzerland; [2]Division of Clinical Chemistry and Biochemistry, University Children's Hospital Zurich, University of Zurich, Zurich, Switzerland; [3]Division of Metabolism and Children's Research Center, University Children's Hospital Zurich, Zurich, Switzerland; [4]Neuroscience Center Zurich, University of Zurich and ETH Zurich, Zurich, Switzerland

**\*For correspondence:**
desiree.boeck@uzh.ch (DB);
patriarchi@pharma.uzh.ch (TP);
schwank@pharma.uzh.ch (GS)

[†]These authors contributed equally to this work

## eLife Assessment

This is an **important** study suggesting that neuron-specific loss of function of the RNA splicing factor Ptbp1 in striatal neurons induces dopaminergic markers and alleviates motor defects in a 6-hydroxydopamine (6-OHDA) mouse model of Parkinson's Disease. The evidence supporting the rescue of motor deficits following Ptbp1 manipulation is **solid**, and, while additional characterization of dopaminergic neuronal identity may be required in future studies, these results have clear implications for Parkinson's disease therapeutics. The study also addresses recent controversial literature on cell reprogramming in Parkinson's Disease and will be of interest to researchers with a focus on the application of gene therapy to rescue neurodegeneration.

**Abstract** Parkinson's disease (PD) is a multifactorial disease caused by irreversible progressive loss of dopaminergic neurons (DANs). Recent studies have reported the successful conversion of astrocytes into DANs by repressing polypyrimidine tract binding protein 1 (PTBP1), which led to the rescue of motor symptoms in a chemically-induced mouse model of PD. However, follow-up studies have questioned the validity of this astrocyte-to-DAN conversion model. Here, we devised an adenine base editing strategy to downregulate PTBP1 in astrocytes and neurons in a chemically-induced PD mouse model. While PTBP1 downregulation in astrocytes had no effect, PTBP1 downregulation in neurons of the striatum resulted in the expression of the DAN marker tyrosine hydroxylase (TH) in non-dividing neurons, which was associated with an increase in striatal dopamine concentrations and a rescue of forelimb akinesia and spontaneous rotations. Phenotypic analysis using multiplexed iterative immunofluorescence imaging further revealed that most of these TH-positive cells co-expressed the dopaminergic marker DAT and the pan-neuronal marker NEUN, with the majority of these triple-positive cells being classified as mature GABAergic neurons. Additional research is needed to fully elucidate the molecular mechanisms underlying the expression of the observed markers and understand how the formation of these cells contributes to the rescue of spontaneous motor behaviors. Nevertheless, our findings support a model where downregulation of neuronal, but not astrocytic, PTBP1 can mitigate symptoms in PD mice.

## Introduction

PD is a complex and multifactorial disorder, characterized by the progressive and irreversible loss of DANs in the substantia nigra pars compacta (SNc), which leads to the disruption of the nigrostriatal pathway and depletion of striatal dopamine (*Bloem et al., 2021*; *Gitler et al., 2017*; *Moore et al., 2005*). The cause of PD is unknown and only a handful of genetic and environmental risk factors have been identified (*Brown et al., 2005*; *de Lau and Breteler, 2006*; *Elbaz et al., 2007*; *Kalia and Lang, 2015*), making the development of a curative therapy challenging. In fact, current treatment strategies do not focus on slowing down or halting disease progression, but rather aim to control symptoms and maintain the patients' quality of life (*Stoker and Barker, 2020*).

Recently emerging *in vivo* transdifferentiation approaches, which leverage the plasticity of specific somatic cell types, hold great promise for developing therapies targeting a wide range of neurode-generative diseases, including PD (*Cohen and Melton, 2011*; *Torper and Götz, 2017*). Astrocytes are of particular interest for such cell fate-switching approaches. First, they are non-neuronal cells and thus not affected by neurodegeneration (*Yu et al., 2020*). Second, they can acquire certain characteristics of neural stem cells, including multipotency, when activated (*Niu et al., 2013*; *Buffo et al., 2008*; *Robel et al., 2011*; *Shimada et al., 2012*; *Sirko et al., 2013*). Finally, they are highly proliferative in brain injuries such as neurodegeneration (*Yu et al., 2021*). Several *in vivo* studies have reported successful reprogramming of astrocytes to neurons via overexpression of proneuronal lineage-specific transcription factors, such as NEUROD1 or SOX2 (*Guo et al., 2014*; *Niu et al., 2015*; *Niu et al., 2013*). Moreover, two recent studies have shown that repression of the RNA-binding protein PTBP1, which mainly functions as a splicing regulator (*Valcárcel and Gebauer, 1997*), efficiently converts astrocytes into DANs in the SNc or striatum (*Qian et al., 2020*; *Zhou et al., 2020*). Consequently, this led to the restoration of the nigrostriatal pathway and striatal dopamine levels, as well as the rescue of motor deficits in a chemically-induced mouse model of PD (*Qian et al., 2020*; *Zhou et al., 2020*). However, since the publication of these two studies in 2020, stringent lineage-tracing strategies have revealed that neither quiescent nor reactive astrocytes convert to DANs upon PTBP1 depletion in the SNc or striatum (*Chen et al., 2022*; *Hoang et al., 2023*; *Wang et al., 2021*), fueling widespread debate about the origin of these *de novo* generated cells and their ability to alleviate motor deficits in PD mice (*Arenas, 2020*; *Jiang et al., 2021*; *Qian et al., 2021*).

In this study, we employed adenine base editors (ABEs), which enable gene editing independent of DNA double-strand break formation (*Gaudelli et al., 2017*) and thus without the risk of inducing chromosomal rearrangements, translocations, or large deletions (*Adikusuma et al., 2018*; *Kosicki et al., 2018*; *Shin et al., 2017*), to install a loss-of-function splice mutation in the *Ptbp1* gene in astrocytes or neurons. Using a chemically-induced PD mouse model, we show that downregulation of neuronal rather than astroglial PTBP1 in the SNc and striatum improves forelimb akinesia and spontaneous rotations. Histological analysis revealed that downregulation of neuronal PTBP1 induced expression of TH, which is the rate-limiting enzyme in the biosynthesis of dopamine and other catecholamines and thus a characteristic marker of DANs, in non-dividing neurons of the striatum. Since lack or dysfunction of TH results in dopamine deficiency and parkinsonism, induction of TH expression in striatal neurons may explain the observed rescue of PD phenotypes in mice and provide therapeutic benefits for PD patients.

## Results

### Adenine base editing effectively downregulates PTBP1 in cell lines

Base editors (BEs) are CRISPR-Cas derived genome engineering tools that allow the precise conversion of A-T to G-C (adenine BEs, ABEs) or C-G to T-A (cytidine BEs, CBEs) base pairs in cell lines as well as post-mitotic cells (*Gaudelli et al., 2017*; *Koblan et al., 2021*; *Komor et al., 2016*; *Levy et al., 2020*; *Villiger et al., 2018*). BEs can thus be applied to precisely disrupt canonical splice sites and permanently eliminate gene function *in vivo* (*Kluesner et al., 2021*; *Musunuru et al., 2021*; *Rothgangl et al., 2021*; *Winter et al., 2019*). To achieve effective and permanent repression of PTBP1, we sought to utilize ABEs to mutate canonical splice sites. To assess if adenine base editing can be used to effectively disrupt PTBP1 expression, we designed seven sgRNAs targeting canonical *Ptbp1* splice donor or acceptor sites in murine Hepa1-6 cells (hereafter referred to as Hepa; *Figure 1—figure supplement 1*). Plasmids expressing the sgRNAs were co-delivered with *Sp*Cas-, *Sp*G-, or

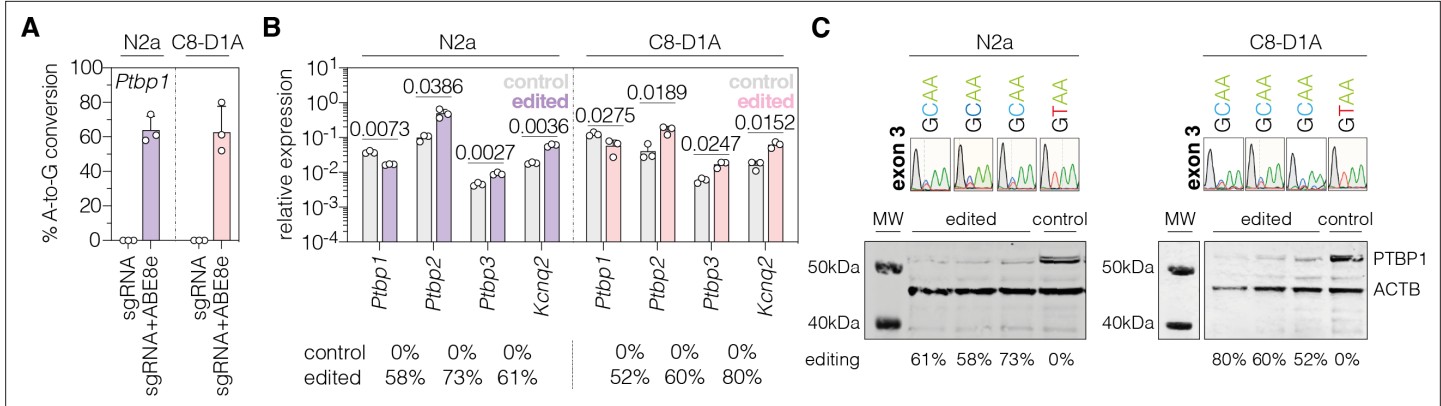

**Figure 1.** Polypyrimidine tract binding protein 1 (PTBP1) downregulation by adenine base editing with sgRNA-ex3 in neuronal and astroglial cell lines. (**A**) Editing rates at the *Ptbp1* splice donor of exon 3 in N2a and C8-D1A cell lines. Editing efficiencies were determined by Sanger sequencing and EditR (***Kluesner et al., 2018***). Control samples were transfected with sgRNA (gray). (**B**) Transcript levels of *Ptbp1* and *Ptbp1*-repressed exons upon adenine base editing in N2a and C8-D1A cells. Transcripts were normalized to *Gapdh*. (**C**) PTBP1 levels in control (1 independent experiment) and edited N2a or C8-D1A cells (3 independent experiments). ACTB protein levels are shown as a loading control. Corresponding sequencing chromatograms for sgRNA-ex3 are shown above each sample. Corresponding editing rates are shown below the plots in (**B**) and (**C**). Normal distribution of the data was analyzed using the Shapiro-Wilk test. Data are represented as means ± s.d. of three independent experiments (**A,B**) and were analyzed using an unpaired two-tailed Student's t-test with Welch's correction (**B**). Each datapoint represents one independent experiment. Exact P-values are indicated in the respective plots. *Ptbp1*, Polypyrimidine tract binding protein 1; *Ptbp2*, Polypyrimidine tract binding protein 2; *Ptbp3*, Polypyrimidine tract binding protein 3; *Kcnq2*, potassium voltage-gates channel subfamily Q member 2; MW, molecular weight marker; kDa, kilodalton.

The online version of this article includes the following source data and figure supplement(s) for figure 1:

**Source data 1.** Original membranes corresponding to ***Figure 1*** (panel C).

**Figure supplement 1.** Adenine base editing of *Ptbp1* splice sites in murine Hepa cells.

**Figure supplement 2.** *In vitro* validation of PTBP1 repression by adenine base editing with sgRNA-ex3 or sgRNA-ex7 in Hepa cells.

**Figure supplement 2—source data 1.** Original membranes corresponding to ***Figure 1—figure supplement 2*** (panel C).

**Figure supplement 3.** Adenine base editing generates alternative polypyrimidine tract binding protein 1 (*Ptbp1*) splice sites in cell lines.

**Figure supplement 3—source data 1.** Original gel images corresponding to ***Figure 1—figure supplement 3*** (panel B).

*Sp*Cas-NG-ABE-expressing plasmids into Hepa cells, and genomic DNA was isolated at 5 d post-transfection for analysis by deep sequencing. In line with previous reports (***Kluesner et al., 2021***), base editing activity was higher at splice donor sites, with the highest editing rates at the exon-intron junctions of exon 3 (92.9 ± 1.0% for *Sp*G-ABE8e) and 7 (85.0 ± 7.2% for *Sp*Cas-ABE8e; ***Figure 1—figure supplement 1***). Next, we validated whether both sgRNAs resulted in a reduction of transcript and protein levels. Average editing rates of 77% (sgRNA-ex3) and 73% (sgRNA-ex7) on genomic DNA (***Figure 1—figure supplement 2***) resulted in approximately 70% (p<0.0004) and 50% reduction of *Ptbp1* transcripts (p<0.0064) in Hepa cells (***Figure 1—figure supplement 2***), leading to a substantial reduction in PTBP1 protein levels and a significant increase in the transcription of exons known to be repressed by PTBP1 (***Han et al., 2014***; ***Li et al., 2014***; ***Figure 1—figure supplement 2***; for sgRNA-ex3: *Ptbp2*, p<0.0014; *Ptbp3*, p<0.0051; *Kcnq2*, p<0.0317; for sgRNA-ex7, *Ptbp2*, p<0.0477; *Ptbp3*, p<0.0055; *Kcnq2*, p<0.0160).

To analyze whether PTBP1 can also be downregulated by adenine base editing in neuronal and astroglial cells, we repeated experiments with sgRNA-ex3 and the ABE8e-*Sp*G variant in the neuronal Neuro2a and astroglial C8-D1A cell lines (hereafter referred to as N2a and C8-D1A). Compared to Hepa cells (92.9 ± 1.0%; ***Figure 1—figure supplement 2***), editing rates were lower in both cell lines (N2a: 64 ± 7.9%; C8-D1A: 62.7 ± 15.1%; ***Figure 1A***). Nevertheless, we again detected a substantial reduction of *Ptbp1* mRNA (N2a, p<0.0073; C8-D1A, P<0.0275) and PTBP1 protein levels (***Figure 1B and C***). Notably, editing of the canonical splice donor at exon 3 generated alternative *Ptbp1* splice sites in all three cell lines (***Figure 1—figure supplement 3***), which, however, did not result in functional PTBP1 protein (***Figure 1C***; ***Figure 1—figure supplement 2***). Based on these results, we decided to use sgRNA-ex3 in combination with the *Sp*G-ABE8e variant for *in vivo* experiments.

## Downregulation of PTBP1 in neurons of the SNc generates TH-expressing cells

To study the effect of PTBP1 downregulation on an injured nigrostriatal circuit in mice, we first induced a unilateral lesion in the medial forebrain bundle (mfb) using the toxic dopamine analogue 6-hydroxydopamine (6-OHDA; *Figure 2—figure supplement 1*; *da Conceição et al., 2010*), similar to previous studies (*Chen et al., 2022*; *Hoang et al., 2023*; *Qian et al., 2020*; *Zhou et al., 2020*). 5 wk after the introduction of a lesion, we quantified the loss of TH$^+$ DANs in the SNc and DA fibers in the striatum by histology (*Figure 2—figure supplement 1*). As expected, 6-OHDA induced a severe unilateral lesion in the nigrostriatal pathway (*Figure 2—figure supplement 1*), characterized by an average 99% reduction in the number of TH$^+$ DANs in the SNc ipsilateral to the injection site (intact hemisphere: 10547 ± 2313 TH$^+$ cells; lesioned hemisphere: 114 ± 83 TH$^+$ cells; *Figure 2—figure supplement 1*) and an average 92% decrease in fluorescence signal corresponding to striatal DA fibers (dorsal, p<0.0001; ventral, p<0.0001; *Figure 2—figure supplement 1*). In line with previous reports (*Chen et al., 2022*), we also observed a sharp increase in activated astrocytes, as indicated by the upregulation of the intermediate filament protein GFAP (glial fibrillary acidic protein; *Figure 2—figure supplement 1*). Finally, we analyzed perturbations in spontaneous motor activities following the 6-OHDA lesion (*Boix et al., 2015*; *Glajch et al., 2012*; *Iancu et al., 2005*) and found that ipsilateral rotations (p=0.0012) and contralateral forelimb akinesia (*P*<0.0001) were significantly increased (*Figure 2—figure supplement 1*).

In order to target PTBP1 in astrocytes or neurons of 6-OHDA-induced PD mice, we designed adeno-associated virus (AAV) vectors expressing the *Sp*G-ABE8e variant under the control of the astrocyte-specific short GFAP promoter (*Lee et al., 2008*) (hereafter referred to as AAV-GFAP), or the neuron-specific human synapsin 1 promoter (hsyn) (*Kügler et al., 2003*) (hereafter referred to as AAV-hsyn). Both vectors additionally express sgRNA-ex3 under the human U6 promoter (*Duvoisin et al., 2012*). As a non-targeting control, we generated an AAV vector that expresses *Sp*G-ABE8e from the ubiquitous Cbh promoter (*Gray et al., 2011*), but does not contain sgRNA-ex3 (hereafter referred to as AAV-ctrl). Since ABE8e exceeds the packaging capacity of a single AAV (~5 kb including ITRs) (*Grieger and Samulski, 2005*), we used the intein-mediated protein trans-splicing system from *Nostoc punctiforme* (*Npu*) (*Li et al., 2008*; *Truong et al., 2015*) to split the ABE for expression from two separate AAVs (*Figure 2—figure supplement 2*). After confirming on-target editing in N2a and C8-D1A cells (*Figure 2—figure supplement 2*), we packaged intein-split ABE8e expression vectors into AAV-PHP.eB capsids and delivered particles to the SNc of C57BL/6 J mice 5 wk after the introduction of the unilateral 6-OHDA lesion (*Figure 2A*). 12 wk after AAV treatment at a dose of 2×10$^8$ vector genomes (vg) per animal, we assessed whether the injured nigrostriatal pathway was reconstituted (*Figure 2A*).

When we first analyzed animals treated with AAV-ctrl, we observed an average 99% reduction of TH$^+$ cells in the SNc of lesioned animals (intact hemisphere: 8678 ± 2765 TH$^+$ cells; lesioned hemisphere: 122 ± 26 TH$^+$ cells; p=0.0331; *Figure 2B and C*). When we next assessed animals treated with AAV-hsyn to downregulate PTBP1 in neurons, we observed a restoration of approximately 10% of TH$^+$ cells compared to the intact hemisphere (intact hemisphere: 7613 ± 1386 TH$^+$ cells; lesioned hemisphere: 721 ± 144 TH$^+$ cells; *Figure 2B and D*). In contrast, when we analyzed animals treated with AAV-GFAP to downregulate PTBP1 in astrocytes, we did not observe TH$^+$ cells above control levels (intact hemisphere: 9209 ± 1199 TH$^+$ cells; lesioned hemisphere: 158 ± 78 TH$^+$ cells; *Figure 2C and D*; *Figure 2—figure supplement 3*). However, despite the presence of TH $^+$ cells in the SNc of AAV-hsyn-treated animals, we did not detect an increase in fluorescence signal corresponding to DA fibers in the striatum (ventral: AAV-GFAP, p=0.7935; AAV-hsyn, p=0.6998; dorsal: AAV-GFAP, p=0.7352; AAV-hsyn, p=0.4906), suggesting that TH$^+$ cells generated in the SNc upon neuronal PTBP1 downregulation did not form striatal projections (*Figure 2E and F*). Further supporting this observation, we did not detect differences in fluorescence intensity between groups when analyzing projections of DANs in the mfb (*Figure 2—figure supplement 4*; AAV-GFAP, p=0.3796; AAV-hsyn, p=0.2996). Notably, base editing at the *Ptbp1* splice site (AAV-ctrl, 0.04 ± 0.03%, AAV-GFAP, 14.7 ± 3.9%; AAV-hsyn, 15.5 ± 8.5%) as well as a reduction of *Ptbp1* transcript (AAV-GFAP, 23.7 ± 12.7%, p=0.0383; AAV-hsyn, 24.8 ± 18.7%, p=0.0354) and PTBP1 protein levels (AAV-GFAP, 10.8 ± 6.3%; AAV-hsyn, 13.8 ± 10.1%) could be confirmed in SNc tissues at experimental endpoints (*Figure 2—figure supplement 5*).

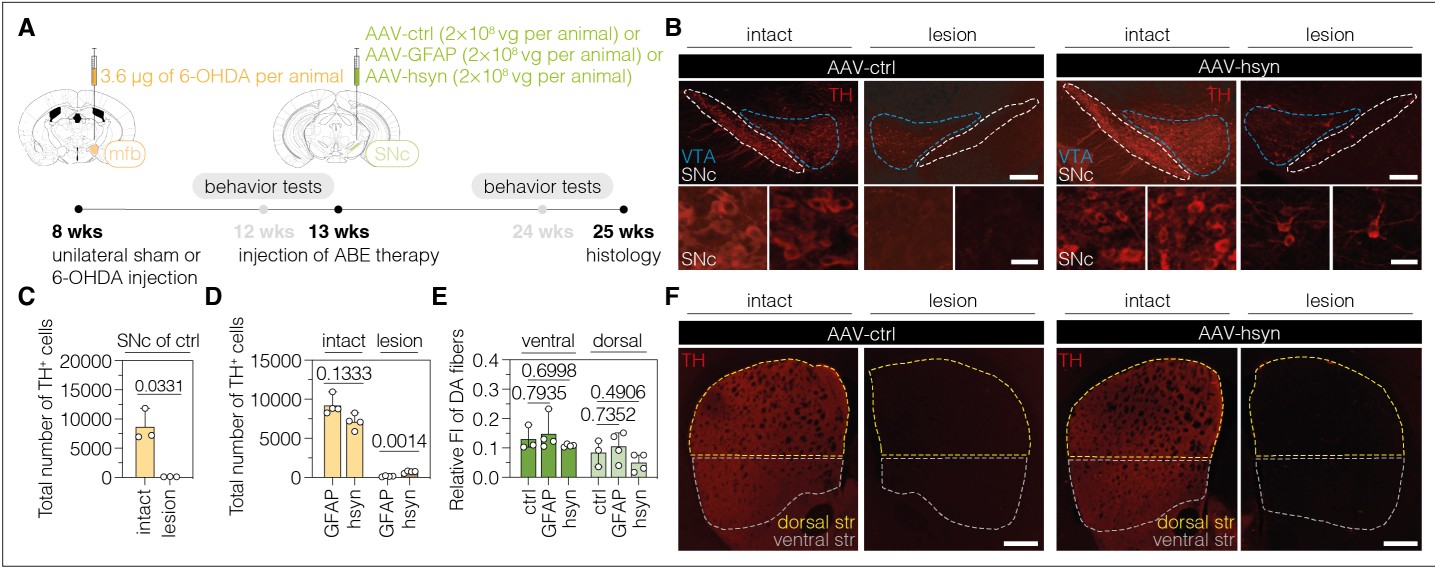

**Figure 2.** Downregulation of polypyrimidine tract binding protein 1 (PTBP1) in neurons of the SNc generates TH[+] cells. (**A**) Schematic representation of the experimental timeline and setup. (**B**) Representative images of midbrain sections showing the intact (left) or lesioned (right) SNc in animals treated with adeno-associated virus (AAV)-ctrl (left) or AAV-hsyn (right). Treatment groups and hemispheres are indicated on top. (**C, D**) Quantification of TH[+] cells in the intact or lesioned SNc in animals treated with AAV-ctrl (**C**), AAV-GFAP (**D**), or AAV-hsyn (**D**) in the lesioned hemisphere. (**E,F**) Quantifications (**E**) of DA fibers in the striatum, assessed as relative fluorescence intensity (FI) of TH compared to the intact striatum of the same section, and representative images of brain sections (**F**) showing the intact or denervated striatum (str) in animals treated with AAV-ctrl (left) or AAV-hsyn (right). The FI of the TH staining detected in the corpus callosum of each hemisphere was used for background correction of FI detected in the striatum of the same hemisphere. Control animals were treated with AAV-PHP.eB particles, expressing the ABE8e variant under the ubiquitous Cbh promoter. Tissue areas used for quantifications are marked by colored dashed lines in (**B**) and (**E**). Normal distribution of the data was analyzed using the Shapiro-Wilk test. Data are represented as means ± s.d. of 3–8 animals per group and were analyzed using an unpaired two-tailed Student's t-test with Welch's correction (**C, D**) or a one-way ANOVA with Dunnett's multiple comparisons test (**E**). Each datapoint represents one animal. Exact p-values are indicated in the respective plots. Scale bars, 20 μm (B, bottom) and 1000 μm (B, top; **F**). ctrl, AAV-ctrl-ABE treatment; GFAP, AAV-GFAP-ABE treatment; hsyn, AAV-hsyn-ABE treatment; ABE, adenine base editor; vg, vector genomes; SNc, substantia nigra pars compacta; VTA, ventral tegmental area; TH, tyrosine hydroxylase; str, striatum; FI, fluorescence intensity; DA, dopaminergic.

The online version of this article includes the following source data and figure supplement(s) for figure 2:

**Figure supplement 1.** Validation of the unilateral 6-OHDA lesion in C57BL/6 J mice.

**Figure supplement 2.** Adeno-associated virus (AAV) vector designs for neuronal or astroglial expression of intein-split ABE8e.

**Figure supplement 3.** Polypyrimidine tract binding protein 1 (PTBP1) downregulation in astrocytes of the SNc fails to generate TH[+] cells.

**Figure supplement 4.** Tyrosine hydroxylase (TH)-expressing cells in the substantia nigra pars compacta (SNc) do not form neuronal projections through the mfb.

**Figure supplement 5.** *In vivo* validation of polypyrimidine tract binding protein 1 (PTBP1) downregulation by adenine base editing in astrocytes and neurons of the SNc.

**Figure supplement 5—source data 1.** Original membranes corresponding to *Figure 2—figure supplement 5* (panel C).

Taken together, our results suggest that the PTBP1 downregulation in neurons of the SNc generates TH[+] cells; however, unlike endogenous DANs in the SNc, they do not project to the dorsal striatum.

## Downregulation of PTBP1 in neurons of the striatum generates TH[+] cells and increases striatal dopamine levels

Since the observed TH[+] cells in the SNc did not generate projections to reconstruct the nigrostriatal pathway, we next tested whether we could bypass the lack of striatal projections by generating TH[+] cells directly in the striatum. We, therefore, delivered AAV-hsyn at a dose of $4×10^8$ vg per animal into the striatum of C57BL/6 J mice, which were pre-treated with 6-OHDA to generate a unilateral lesion. The injection volume, and thus the delivered AAV dose, was increased compared to the SNc to achieve comparable AAV biodistribution in the larger striatum. Confirming the unilateral impairment of the nigrostriatal pathway, analysis of brain sections at 12 wk post-treatment revealed an average

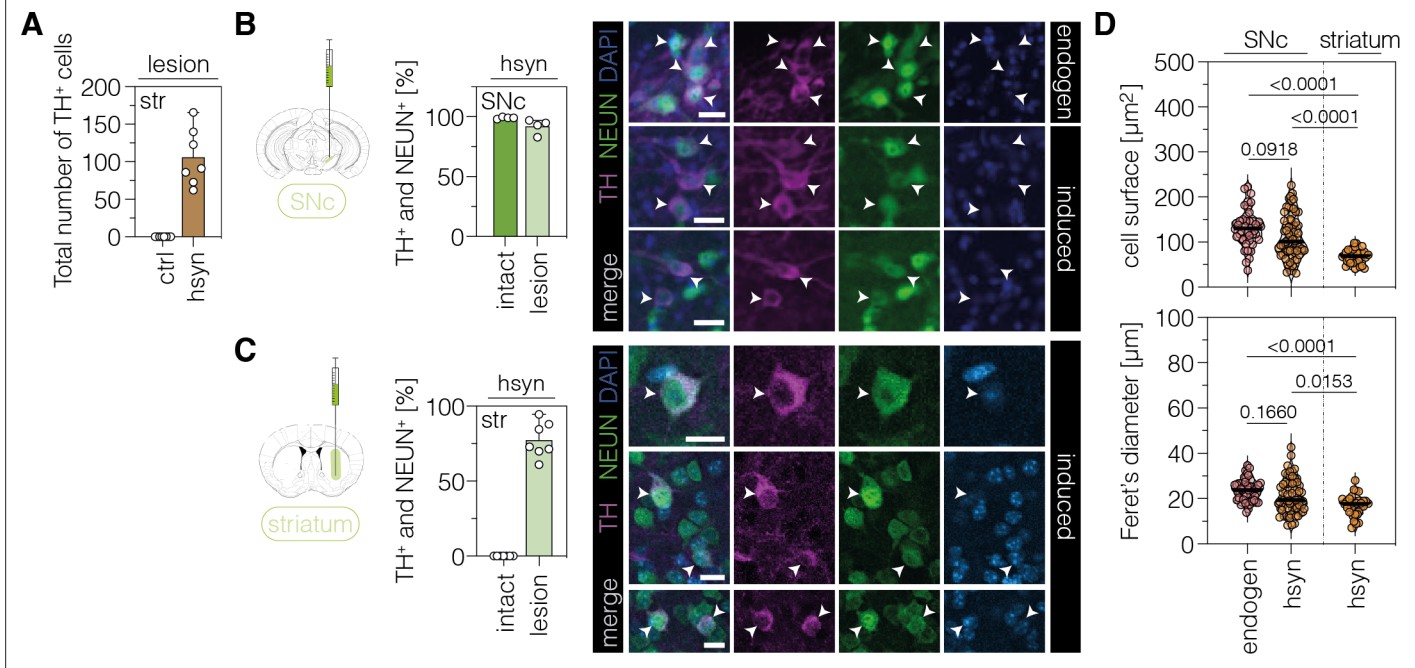

**Figure 3.** Characterization of TH-expressing cells in the SNc or striatum. (**A**) Quantification of TH⁺ cells in the lesioned striatum of animals treated with AAV-ctrl or AAV-hsyn. Control animals were treated with AAV-PHP.eB particles, expressing the ABE8e-*Sp*G variant under the ubiquitous Cbh promoter. (**B,C**) Quantifications (left) and representative images (right) of TH/NeuN double-positive cell bodies in the intact (dark green, labeled as 'endogen' in the images) or lesioned (light green, labeled as 'induced' in the images) SNc (**B**) or striatum (**C**) of AAV-hsyn-treated animals. (**D**) Corresponding quantifications of cell surface area and Feret's diameter (longest distance between the cell boundaries) of TH/NeuN double-positive cell bodies in the intact (labeled as 'endogen') or lesioned SNc or striatum (labeled as 'hsyn'). Normal distribution of the data was analyzed using the Shapiro-Wilk test. Data are displayed as means ± s.d. of 4–7 mice per group (**A–C**) or 32–68 TH/NeuN double-positive cells per group (D; n=4 mice) and were analyzed using a one-way ANOVA with Tukey's multiple comparisons test (**D**). Each datapoint represents one animal (**A–C**) or a TH/NeuN double-positive cell (**D**). Exact P-values are indicated in the each plot above the respective group (**D**). Scale bars, 20 μm. ctrl, AAV-ctrl-ABE treatment; hsyn, AAV-hsyn-ABE treatment; SNc, substantia nigra pars compacta; str, striatum; TH, tyrosine hydroxylase; NEUN, hexaribonucleotide binding protein-3; DAPI, 4',6-diamidino-2-phenylindole; endogen, endogenous.

The online version of this article includes the following source data and figure supplement(s) for figure 3:

**Figure supplement 1.** Validation of the unilateral lesion in 6-OHDA mice after neuronal polypyrimidine tract binding protein 1 (PTBP1) downregulation in the striatum.

**Figure supplement 2.** *In vivo* validation of polypyrimidine tract binding protein 1 (PTBP1) downregulation by adenine base editing in neurons of the striatum.

**Figure supplement 2—source data 1.** Original membranes corresponding to *Figure 3—figure supplement 2* (panel D).

**Figure supplement 3.** Identification of sgRNA-dependent off-target sites using GUIDE-seq.

**Figure supplement 4.** Neuronal polypyrimidine tract binding protein 1 (PTBP1) repression in the striatum increases striatal dopamine in 6-OHDA-lesioned hemispheres.

**Figure supplement 5.** Absence of TH⁺ cell bodies in the visual cortex after neuronal polypyrimidine tract binding protein 1 (PTBP1) downregulation.

**Figure supplement 6.** Absence of proliferation in TH⁺ cell bodies in the striatum after neuronal polypyrimidine tract binding protein 1 (PTBP1) downregulation.

99% reduction of TH⁺ cells in the lesioned SNc and no detectable DA projections to the striatum (*Figure 3—figure supplement 1*). When we quantified TH⁺ cells in brain sections of the striatum, we found 106±38 TH⁺ cells across 3 sections (estimated as 2646±952 cells in the entire striatum) in mice treated with AAV-hsyn compared to no TH⁺ cells in the lesioned hemisphere of animals treated with AAV-ctrl (*Figure 3A*). Analysis of striatum tissues isolated from AAV-ctrl- and AAV-hsyn-treated PD mice revealed base editing at the *Ptbp1* splice site (AAV-ctrl, 0.4 ± 0.1%; AAV-hsyn, 23.0 ± 6.6%; *Figure 3—figure supplement 2*) and downregulation of *Ptbp1* transcript (AAV-hsyn, 30.8 ± 14.6%, *P*=0.0681) and PTBP1 protein levels (AAV-hsyn, 22.1 ± 6.4%; *Figure 3—figure supplement 2*).

Since we did not observe PTBP1 downregulation or TH$^+$ cells in animals treated with AAV-ctrl (*Figure 3A*; *Figure 3—figure supplement 2*), we can rule out that TH expression is induced by (1) tissue damage due to the injection procedure or (2) toxicity due to the administered AAV dose or serotype. To additionally exclude that TH expression might be caused by off-target editing effects, we experimentally determined off-target sites of sgRNA-ex3 using GUIDE-seq in N2a cells (*Tsai et al., 2015*) and subsequently analyzed these sites in treated animals by deep amplicon sequencing (*Figure 3—figure supplement 3*). One off-target site was identified in the myopalladin (*Mypn*) gene, which encodes for a muscle-specific protein and plays a critical role in regulating the structure and growth of skeletal and cardiac muscle (*Filomena et al., 2021*; *Filomena et al., 2020*), and another site was detected in the intronic region of the ankyrin-1 (*Ank1*) gene, which encodes for an adaptor protein linking membrane proteins to the underlying cytoskeleton (*Cunha and Mohler, 2009*). Importantly, base editing rates at both sites were substantially lower (*Mypn*, 5.4 ± 1.7%; *Ank1*, 5.8±1.9%; *Figure 3—figure supplement 3*) than at the *Ptbp1* site in AAV-hsyn-treated mice (23.0 ± 6.6%; *Figure 3—figure supplement 2*) and no reduction in transcript levels of the respective genes was observed (*Mypn*, P=0.4528; *Ank1*, P=0.5547; *Figure 3—figure supplement 3*). Thus, the induction of TH expression upon adenine base editing with sgRNA-ex3 is likely a direct consequence of PTBP1 downregulation.

Next, we assessed whether the ectopic expression of TH in the striatum led to an increase in tissue dopamine levels. We manually dissected striata of lesioned and unlesioned hemispheres and quantified tissue dopamine levels by high-pressure liquid chromatography (HPLC; *Figure 3—figure supplement 4*). Peak identities and neurotransmitter concentrations were analyzed using corresponding standards of known concentrations. Importantly, base-edited animals showed an approximately 2.5-fold increase in the concentration of striatal dopamine (AAV-hsyn: 4.4 ± 2.6 nmol/g protein; AAV-ctrl: 1.7 ± 0.7 nmol/g protein; p=0.0159) and the dopamine metabolite 3,4-dihydroxyphenylacetic acid (DOPAC; AAV-hsyn: 8.2 ± 3.9 nmol/g protein; AAV-ctrl: 3.1 ± 1.8 nmol/g protein; p=0.0449) in the lesioned hemisphere (*Figure 3—figure supplement 4*).

Taken together, our data suggest that downregulation of PTBP1 in striatal neurons of 6-OHDA-lesioned hemispheres resulted in the expression of TH and an elevation of striatal dopamine concentrations.

## Phenotypic characterization of TH$^+$ cells in the striatum

To characterize the TH-expressing cells in the SNc and striatum of AAV-hsyn-treated mice in more detail, we co-stained them for the pan-neuronal marker NEUN (hexaribonucleotide binding protein-3). As expected, virtually all TH$^+$ cells in the intact SNc were co-stained for NEUN (99.0 ± 0.1%; *Figure 3B*). Likewise, the majority of TH$^+$ cells in the lesioned SNc (hsyn: 92.0 ± 6.2%; *Figure 3B*) and striatum of AAV-hsyn-treated animals (77.5 ± 11.8%; *Figure 3C*) were also labeled by NEUN, further corroborating a neuronal origin of these cells. Moreover, we observed a significant reduction in the surface area (p<0.0001) and Feret's diameter (p<0.0001) between TH/NEUN double-positive cells in the lesioned striatum compared to endogenous DANs in the intact SNc, or TH/NEUN double-positive cell bodies in the lesioned SNc of AAV-hsyn-treated mice (*Figure 3D*; surface area, p<0.001; Feret's diameter, p<0.001).

Differences in the frequency of NEUN-labeling, as well as surface area and diameter of TH$^+$ cells in the striatum, might be attributed to (1) varying maturation states of these cells, potentially influenced by the local microenvironment of the SNc vs striatum, and/or (2) TH-expressing cells originating from distinct neuronal subpopulations. In line with previous work (*Qian et al., 2020*), injection of AAV-hsyn into the visual cortex did not lead to TH expression in local neurons (*Figure 3—figure supplement 5*; n=174 Cas9/NEUN double-positive cells), suggesting that the local microenvironment indeed plays a contributing role in inducing TH expression in the targeted neurons. To next analyze the origin of TH-expressing cells in the striatum, we first assessed whether these cells originated from dividing neural progenitors or mature, non-dividing neurons. We, therefore, supplied bromodeoxyuridine (BrdU)-containing drinking water to 6-OHDA-lesioned mice after AAV-hsyn treatment (*Figure 3—figure supplement 6*). After confirming the successful BrdU labeling of proliferating cells in the dentate gyrus (DG; *Figure 3—figure supplement 6*), we performed BrdU/NEUN/TH co-staining experiments with striatal sections. Microscopic analysis of 163 TH/NEUN

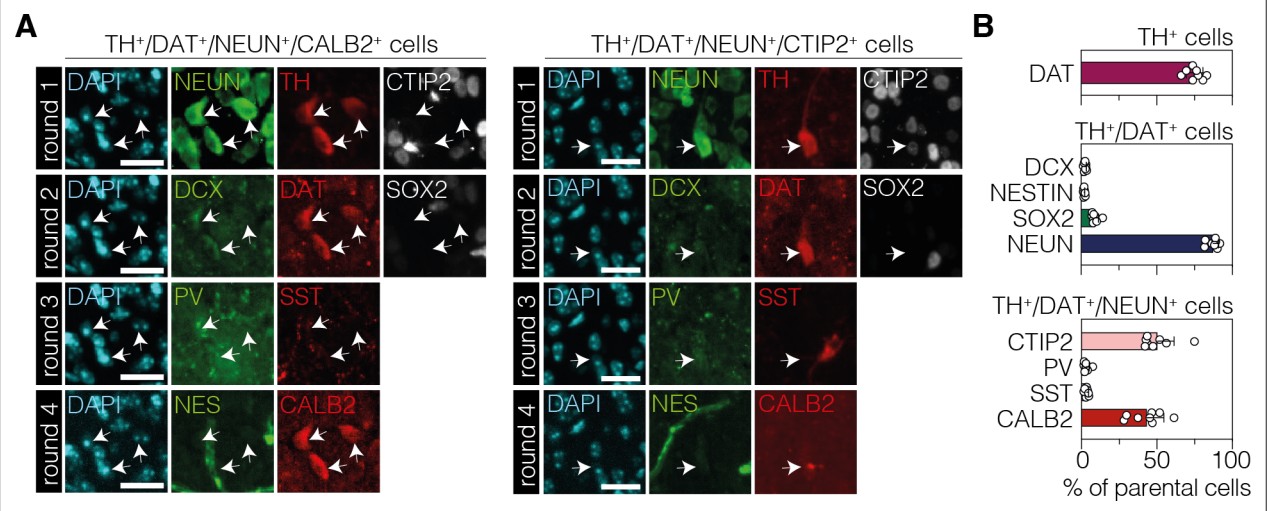

**Figure 4.** Characterization of TH⁺ cells in the striatum after neuronal polypyrimidine tract binding protein 1 (PTBP1) downregulation. (**A**) Representative 4i images of the two main cell populations among TH-positive cells. The expressed markers are indicated on top of the images. (**B**) Corresponding quantifications of phenotypic markers expressed among TH-positive cells (top, DAT; n=696 TH⁺ cells), TH/DAT double-positive (middle, DCX; NESTIN; SOX2; n=527 cells), or TH/DAT/NEUN triple-positive (bottom, CTIP2; PV; SST; CALB2; n=460 cells) subpopulations (white arrows) in the striatum at 12 wk after administration of the AAV-hsyn treatment. The parental population is indicated above each plot. 4i imaging rounds are indicated on the left in (**A**). Images were pseudocolored during post-processing. Scale bars, 20 µm. Data are displayed as means of six mice (**B**). DAPI, 4′,6-diamidino-2-phenylindole; SOX2, sex determining region Y-box 2; NES, neuroepithelial stem cell protein; DCX, doublecortin; NEUN, hexaribonucleotide binding protein-3; TH, tyrosine hydroxylase; DAT, dopamine transporter; CTIP2, COUP-TF-interacting protein 2; PV, parvalbumin; SST, somatostatin; CALB2, calbindin 2.

The online version of this article includes the following figure supplement(s) for figure 4:

**Figure supplement 1.** Validation of antibodies for 4i experiments.

double-positive cells revealed no co-labeling with BrdU (*Figure 3—figure supplement 6*), indicating that TH⁺ cells were not generated *de novo* and rather originated from non-proliferating mature neurons.

Next, we performed multiplexed iterative immunofluorescence imaging (4i) (*Cole et al., 2022*) on tissue sections to further characterize the identity, origin, and differentiation state of these cells (*Figure 4*). After successful validation of antibody specificities (*Figure 4—figure supplement 1*), we performed four 4i rounds using markers for neural progenitors (SOX2/sex-determining region Y-box 2, NES/neuroepithelial stem cell protein, DCX/doublecortin), DANs (TH, DAT), and mature neurons (NEUN, CTIP2/COUP-TF-interacting protein 2, SST/somatostatin, PV/parvalbumin, CALB2/calbindin 2). The majority of TH⁺ cells also expressed the marker DAT (75.2 ± 5.6%; *Figure 4A and B*), further corroborating the DA identity of these cells. Moreover, supporting the results of our BrdU-labeling experiments (*Figure 3—figure supplement 6*), only a small fraction of TH/DAT double-positive cells were labeled for markers of neural progenitors (SOX2, 9.1 ± 2.4%; NESTIN, 2.1 ± 0.5%; DCX, 2.6 ± 0.8%; *Figure 4A and B*). Instead, most TH/DAT-labeled cells expressed the adult pan-neuronal marker NEUN (86.2 ± 3.7%; *Figure 4A and B*). Of this TH/DAT/NEUN-positive population, 49.8 ± 12.9% were additionally labeled with CTIP2 (*Figure 4A and B*), a marker for GABAergic medium spiny neurons (MSNs), and 46.4 ± 10.4% expressed markers for various GABAergic interneurons (PV, 3.2 ± 1.9%; SST, 3.5 ± 1.2%; CALB2, 39.7 ± 11.0%; *Figure 4A and B*), indicating that expression of DA markers may be achieved in various subtypes of GABAergic neurons upon PTBP1 downregulation.

In summary, our data show that TH-expressing cells were not generated from proliferating neural stem cells, but rather originated from various populations of post-mitotic striatal neurons that acquired DA characteristics upon PTBP1 downregulation.

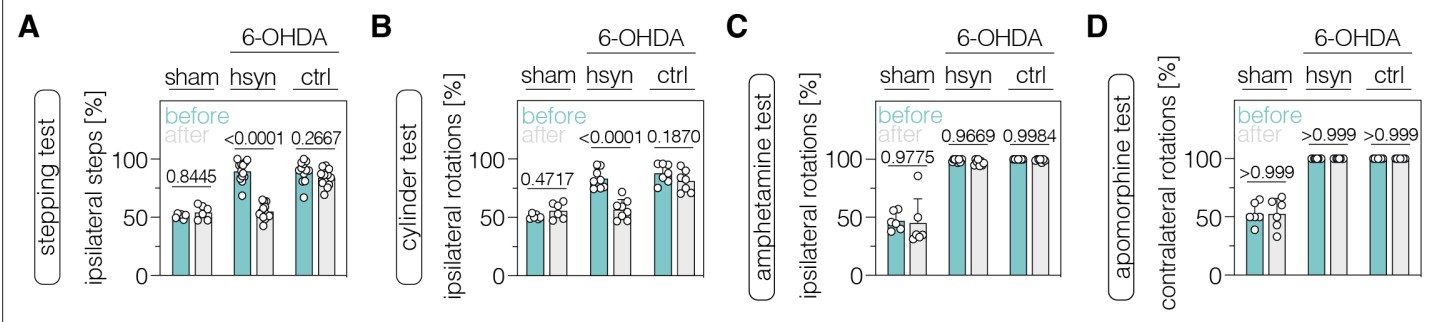

**Figure 5.** Neuronal polypyrimidine tract binding protein 1 (PTBP1) downregulation in the striatum alleviates drug-free motor dysfunction in 6-OHDA-lesioned Parkinson's disease (PD) mice. (**A, B**) Spontaneous behaviors, were assessed as contralateral forelimb akinesia in the stepping test (**A**) and spontaneous rotations in the cylinder test (**B**), in animals treated in the striatum. (**C, D**) Drug-induced rotations, assessed as amphetamine-induced ipsilateral rotations (**C**) and apomorphine-induced contralateral rotations (**D**), in animals treated in the striatum. Normal distribution of the data was analyzed using the Shapiro-Wilk test. Data are represented as means ± s.d. of 6–13 animals per group and were analyzed using a two-way ANOVA with Šidák's multiple comparisons. Each datapoint represents one animal. Exact p-values are indicated in each plot. hsyn, AAV-hsyn-ABE treatment; ctrl, AAV-ctrl-ABE treatment.

The online version of this article includes the following figure supplement(s) for figure 5:

**Figure supplement 1.** Neuronal polypyrimidine tract binding protein 1 (PTBP1) downregulation in the substantia nigra pars compacta (SNc) alleviates drug-free motor dysfunction in 6-OHDA-lesioned PD mice.

**Figure supplement 2.** Schematic representation of the polypyrimidine tract binding protein 1 (PTBP1)/nPTBP regulatory loops driving neuronal differentiation and maturation.

## Neuronal PTBP1 repression alleviates drug-free motor dysfunction in PD mice

Last, we evaluated whether neuronal and/or astroglial base editing of PTBP1 in the SNc and/or striatum could restore motor functions in mice with a unilateral 6-OHDA lesion. We first performed two common drug-free behavioral tests: the cylinder test to quantify the asymmetry of spontaneous rotations and the stepping test to quantify contralateral forelimb akinesia (*Boix et al., 2015*; *Glajch et al., 2012*; *Iancu et al., 2005*). We found that drug-free motor dysfunctions were significantly alleviated in animals treated with AAV-hsyn (stepping test, p<0.0001; cylinder test, p<0.0001), but not with AAV-GFAP (stepping test, p=0.8608; cylinder test, p=0.0504), in the SNc when using an AAV dose of 2×10[8] vg per animal (*Figure 5—figure supplement 1*). Likewise, PTBP1 targeting in striatal neurons restored the asymmetry of spontaneous behaviors (AAV-hsyn: stepping test, p<0.0001; cylinder test, p<0.0001; AAV-ctrl: stepping test, p=0.2667; cylinder test, p=0.1870; *Figure 5A and B*). To assess the extent of motor improvements in response to the ABE treatment, we additionally tested two drug-induced motor behaviors. However, we did not detect recovery of contralateral rotations in treated animals after systemic administration of amphetamine (*Figure 5C*; *Figure 5—figure supplement 1*), which leads to an enhanced imbalance of extracellular dopamine concentrations between the denervated and intact striatum (*Freyberg et al., 2016*; *Karam et al., 2022*). Likewise, after systemic administration of apomorphine, which acts as a dopamine receptor agonist and stimulates hypersensitive dopamine receptors in the lesioned hemisphere (*Arroyo-García et al., 2018*; *da Conceição et al., 2010*; *Iancu et al., 2005*), we did not observe a recovery of ipsilateral rotations in any treatment group (*Figure 5D*; *Figure 5—figure supplement 1*).

Taken together, downregulation of PTBP1 in SNc and striatal neurons improves spontaneous, but not drug-induced, behaviors in 6-OHDA-lesioned mice.

## Discussion

In this study, we applied adenine base editing to introduce *Ptbp1* loss-of-function mutations in astrocytes and neurons in the 6-OHDA-induced PD mouse model. Delivery of dual AAV vectors to the SNc resulted in the formation of TH-expressing cells and the rescue of spontaneous behaviors when a neuronal promoter was used to drive ABE expression. However, no DA projections to the striatum were detected, supporting recent findings that suggest a functional role of PTBP1 in promoting

axon regeneration of adult sensory neurons (*Alber et al., 2023*). Reconstitution of the nigrostriatal pathway, which connects the SNc with the dorsal striatum (*Kalia and Lang, 2015*), is therefore unlikely the mechanism underlying the observed phenotypic rescue. Supporting this hypothesis, the downregulation of neuronal PTBP1 in the striatum of 6-OHDA-lesioned mice also led to the formation of TH$^+$ cells and a rescue of spontaneous behaviors.

Two previous studies suggested that PTBP1 downregulation in the SNc, using either shRNA-mediated knockdown or knockdown via CRISPR-CasRx, led to the conversion of astrocytes into functional DANs in 6-OHDA-lesioned mice (*Qian et al., 2020*; *Zhou et al., 2020*). However, recent lineage tracing studies revealed that neither quiescent nor reactive astrocytes in the SNc or striatum convert to DANs upon PTBP1 downregulation (*Chen et al., 2022*; *Hoang et al., 2023*; *Wang et al., 2021*). Instead, these studies hypothesize that the reported effects might be attributed to leaky activation of the GFAP promoter in neurons, which could have been misinterpreted as astrocyte-to-neuron conversion. Our study contributes to this recently growing body of evidence, indicating that downregulation of astroglial PTBP1 does not induce astrocyte conversion into DANs, and that neurons are the origin of the observed TH$^+$ cells (*Chen et al., 2022*; *Wang et al., 2021*; *Yang et al., 2023*). Our 4i and BrdU experiments furthermore revealed that TH expression in the striatum was induced in mature neurons, either expressing markers of GABAergic MSNs or interneurons. These findings indicate that the induction of TH expression is not restricted to a specific neuronal subpopulation, but may rather be attributed to (1) potential bias of hsyn promoter activity towards inhibitory neurons (*Radhiyanti et al., 2021*), (2) preferential AAV tropism towards distinct neuronal subpopulations in the striatum, or (3) differences in the local microenvironment, chemical signals, and neuronal circuitry required to induce TH expression in distinct neuronal subpopulations.

While our study suggests that PTBP1 downregulation can enable TH expression in various GABAergic neuronal populations in the striatum, the transcriptional changes leading to and following TH expression in these cells remain unclear. Previous research has demonstrated that the interplay between microRNAs (miRNAs) and the RNA processing proteins PTBP1 and its homology nPTBP (PTBP2) is crucial during neuronal differentiation and maturation (*Figure 5—figure supplement 2*; *Boutz et al., 2007*; *Makeyev et al., 2007*; *Zheng et al., 2012*). Key events in this signaling cascade are the dynamic release of PTBP1-mediated inhibition of miRNA-124, which induces nPTBP expression and activates neuron-specific expression programs in non-neuronal cells (*Makeyev et al., 2007*; *Xue et al., 2013*). Thus, the activation of the PTBP1/nPTBP regulatory loops is essential for driving reprogramming toward the neuronal lineage and subsequent neuronal maturation (*Xue et al., 2016*; *Xue et al., 2013*). While this classical model of PTBP1 down-regulation and neuronal differentiation implies negligible expression or importance of PTBP1 in mature neurons, our results indicate that PTBP1 is sufficiently expressed and plays a functional role in mature neurons, with the installation of a loss-of-function mutation resulting in a detectable reduction in relative protein amounts (*Figure 2—figure supplement 5*; *Figure 3—figure supplement 2*) and behavioral improvements in a PD mouse model (*Figure 5*; *Figure 5—figure supplement 1*). Supporting these findings, a recent study demonstrated PTBP1 expression in axons of sensory and motor neurons as well as a functional role in sensation, injury response, and axonal regeneration (*Alber et al., 2023*). Further investigation using single cell transcriptional analysis of the PTBP1/nPTBP regulatory loops, pro-neuronal transcription factors (*Ascl1*, *Myt1l*, *Zic1*, *Brn2*, *Neurod1*) (*Xue et al., 2013*), or transcription factors guiding the differentiation into DANs (*Nr4a2*, *Lmx1a*, *Lmx1b*, or *Pitx3 Niu et al., 2021*) could provide insights into the transcriptional switches underlying TH expression in MSNs and interneurons as observed in our study.

MSNs, the principal neurons of the striatum (~95%), can be divided into two distinct subtypes based on their expression of dopamine receptors and their axonal projections (D1- or D2-MSNs) (*Gerfen et al., 1990*; *Anonymous, 1998*). Both the input to and output from D1- and D2-MSNs are dynamically controlled by striatal interneurons, which constitute approximately 5% of total striatal neurons (*Clarke and Adermark, 2015*). Since both MSNs and interneurons receive DA inputs from the SNc, their activity, and excitability are strongly hampered when striatal dopamine is absent (*Bamford et al., 2018*; *Gerfen and Surmeier, 2011*; *Kreitzer and Malenka, 2008*). It, therefore, seems feasible that PTBP1 downregulation by adenine base editing and subsequent TH expression might have enabled local dopamine synthesis in the dorsal striatum, which may have been sufficient to partly restore the function of these cell populations and compensate for the depletion of DA inputs from the SNc. Supporting this hypothesis, a recent study has shown that depletion of the

orphan G-protein coupled receptor 6 (GPR6) in D2-MSNs reduced intracellular cyclic adenosine monophosphate (cAMP) concentrations, leading to increased striatal dopamine and decreased involuntary movements following apomorphine treatment in 6-OHDA-lesioned PD mice (*Sun et al., 2021*). When we quantified dopamine levels in the striatum of AAV-hsyn-treated PD mice using HPLC, we observed a 2.5-fold increase in striatal dopamine, suggesting that induction of TH expression in striatal neurons might establish a basal tone of extracellular dopamine despite the absence of striatal DA projections, akin to pharmacological replenishment in PD patients (*Poewe et al., 2017*). While the increase in dopamine levels was modest compared to 6-OHDA-lesioned rats treated with the dopamine precursor L-DOPA, which exhibited an increase in extracellular dopamine concentrations from ~0.04 fmol/µL to ~9–10 fmol/µL (*Lindgren et al., 2010*), our results indicate that even a modest increase in striatal dopamine is sufficient to rescue spontaneous motor behaviors in PD mice. This finding aligns with recent work showing that multiple unconditioned DA-dependent motor tasks can be sustained through a diffuse basal tone of extracellular dopamine in the striatum (*Delignat-Lavaud et al., 2023*). Conversely, basal dopamine levels were insufficient to improve amphetamine-induced behaviors in AAV-hsyn-treated PD mice in this study, potentially due to the lack of striatal projections from the SNc and/or insufficient availability of intracellular dopamine in striatal neurons. Likewise, the detected dopamine levels in striatal tissues were not sufficient to reverse the hypersensitivity of D1 and D2 receptors to the dopamine agonist apomorphine. Higher base editing rates, achievable using either single AAV delivery of smaller Cas9 orthologues or multiple injections into the rostral, medial, and caudal regions of the striatum, may result in more pronounced global PTBP1 downregulation and potentially higher tissue dopamine concentrations, leading to a broader behavioral rescue. Moreover, further analysis using fiber photometry or microdialysis could provide information on synaptic dopamine release and availability.

Transcriptional reprogramming is fundamental for neuronal differentiation and maturation, relying on various feedback and feed-forward circuits to regulate cell type-specific gene expression programs. While the loss of PTBP1 is considered crucial for reducing neuronal apoptosis and inducing neuronal differentiation during development, its various roles in mature neurons during adulthood remain largely elusive. Further investigation using single-cell transcriptional analysis or spatial transcriptomics is needed to identify the transcriptional changes driving the expression of DA markers in striatal GABAergic neurons upon PTBP1 downregulation. These experiments may also contribute to our understanding of how PTBP1 downregulation may affect cAMP metabolism, dopamine synthesis, and basal ganglia circuitry in PD mice. Finally, to fully explore the therapeutic potential of PTBP1 base editing, future studies should also combine detailed transcriptional analysis with stringent lineage-tracing technologies and *in vivo* assessment of synaptic dopamine availability and release in more sophisticated PD mouse models that exhibit neuropathological, motor, and cognitive aspects of the disease.

## Materials and methods
### Generation of plasmids
sgRNA plasmids were generated by ligating annealed and phosphorylated oligos into a BsmBI-digested lentiGuide-Puro (Addgene #52963) using T4 DNA ligase (NEB). To generate intein-split ABE plasmids for AAV production, inserts with homology overhangs were either ordered as gBlocks (IDT) or generated by PCR. Inserts were cloned into KpnI- and AgeI-digested AAV backbones using HiFi DNA Assembly Master Mix (NEB). All PCRs were performed using Q5 High-Fidelity DNA Polymerase (NEB). All plasmids were transformed into *Escherichia coli* Stable3 competent cells (NEB). The identity of all plasmids was confirmed by Sanger Sequencing. Primers used for cloning of all plasmids are listed in *Supplementary files 1 and 2*. LentiGuide-Puro was a gift from F. Zhang (Addgene plasmid nos. 52963).

### Cell culture transfection and genomic DNA preparation
Hepa1-6 (ATCC CRL-1830) cells were maintained in Dulbecco's modified Eagle's medium (DMEM) plus GlutaMAX (Thermo Fisher Scientific), supplemented with 10% (v/v) fetal bovine serum (FBS) and 1% penicillin/streptomycin (Thermo Fisher Scientific) at 37 °C and 5% $CO_2$. Neuro2a (ATCC CCL-131) cells were maintained in Eagle's Minimum Essential Medium (EMEM), supplemented with 10% (v/v) FBS

and 1% penicillin/streptomycin. C8-D1A [astrocyte type I clone, ATCC CRL-2541] were maintained in Dulbecco's modified Eagle's medium (DMEM) supplemented with 10% (v/v) fetal bovine serum (FBS) and 1% penicillin/streptomycin (Thermo Fisher Scientific) at 37 °C and 5% $CO_2$. Cells were passaged every 3–4 d and maintained at confluency below 90%. The identity of all cell lines was confirmed by STR profiling at ATCC. All cell lines were tested negative for mycoplasma contamination.

For the *in vitro* screening of sgRNA activities, cells were seeded in 96-well cell culture plates (Greiner) and transfected at 70% confluency using 0.5 µl Lipofectamine 2000 (Thermo Fisher Scientific). If not stated otherwise, 300 ng of BE and 100 ng of sgRNA were used for transfections. Cells were incubated for 5 d after transfection and genomic DNA was isolated using a direct lysis as previously described (*Böck et al., 2022*). For analysis of transcript and protein levels, cells were seeded in a 48-well cell culture plate (Greiner) and transfected at 70% confluency using 1 µl of Lipofectamine 2000 (Thermo Fisher Scientific). A small aliquot of the cells was used for the isolation of genomic DNA by direct lysis as previously described (*Böck et al., 2022*). The remaining cells were split in half for RNA and protein isolation.

For the assessment of *in vivo* base editing performance, a 40 µm thick section of the SNc or striatum was used for manual dissection of these regions under a stereomicroscope. DNA was subsequently isolated from tissue pieces by direct lysis as previously described (*Böck et al., 2022*).

## RNA isolation and RT-qPCR

RNA was isolated from cultured cells or snap-frozen brain tissues (striatum or SNc) using the RNeasy Mini Kit (Qiagen) or the AllPrep DNA/RNA/Protein Mini Kit (Qiagen) according to the manufacturer's instructions. RNA (1000 ng input) was subsequently reverse-transcribed to cDNA using random primers and the GoScript reverse transcriptase kit (Promega). RT-qPCR was performed using FIRE-Polymerase qPCR Master Mix (Solis BioDyne) and analyzed using a Lightcycler 480 system (Roche). Fold changes were calculated using the double ΔCt method. Primers used for RT-qPCR are listed in *Supplementary file 3*.

## Protein isolation and western blot

Protein was isolated from cultured cells or snap-frozen brain tissues (striatum or SNc) using radioimmunoprecipitation (RIPA) assay buffer (150 mM Tris pH 8.0, 150 mM NaCl, 0.1% SDS, 0.5% sodium deoxycholate, 1% NP-40; Thermo Fisher Scientific), supplemented with protease inhibitor cocktail (Roche), or the AllPrep DNA/RNA/Protein Mini Kit (Qiagen) according to the manufacturer's instructions. Protein concentrations of all samples were determined using the Pierce Bicinchoninic Acid (BCA) Protein Assay Kit (Thermo Fisher Scientific).

Equal amounts of protein (*in vitro* samples: 30 µg; *in vivo* samples: 40 µg) were separated by SDS-polyacrylamide gel electrophoresis (Thermo Fisher Scientific) and transferred to a 0.45 µm nitrocellulose membrane (Amersham). Membranes were incubated with rabbit anti-PTBP1 (1:10,000; cat. no. ab133734, Abcam) and mouse anti-actin beta (1:2,000; cat. no. ab8226; Abcam). Signals were detected by fluorescence using IRDye-conjugated secondary antibodies (LI-COR Biosciences) and a LICOR Odyssey DLx imaging system. Protein quantifications were performed in Fiji. All antibodies are listed in *Supplementary file 4*.

## Amplification for deep sequencing

PTBP1-specific oligos were used to generate targeted amplicons for deep sequencing. Input genomic DNA was first amplified in a 10 µL reaction for 30 cycles using NEBNext High-Fidelity 2x PCR Master Mix (NEB). Amplicons were purified using AMPure XP beads (Beckman Coulter) and subsequently amplified for eight cycles using oligos with sequencing adapters. Approximately equal amounts of PCR products were pooled, gel purified, and quantified using a Qubit 3.0 fluorometer and the dsDNA HS Assay Kit (Thermo Fisher Scientific). Paired-end sequencing of purified libraries was performed on an Illumina Miseq platform. Oligos for deep sequencing are listed in *Supplementary file 5*.

## HTS data analysis

Sequencing reads were first demultiplexed using the Miseq Reporter (Illumina). Next, amplicon sequences were aligned to their reference sequences using CRISPResso2 (*Clement et al., 2019*). Adenine base editing efficiencies at splice sites were calculated as a percentage of (number of reads

containing edits at splice site)/(number of total aligned reads). Reference nucleotide sequences are listed in **Supplementary file 6**.

## GUIDE-seq off-target analysis

Briefly, $6×10^4$ N2a cells were plated in a 24-well plate 1 d prior to transfection. Cells were transfected using Lipofectamine 3000 according to the manufacturer's instructions. Three plasmids, encoding for the *Sp*G Cas9 nuclease [333 ng], the *Ptbp1* sgRNA-3 [167 ng], and GFP [6 ng], were co-transfected with 12 pmol of the double-stranded oligodeoxynucleotide (dsODN) following the original GUIDE-seq protocol (*Tsai et al., 2015*). Transfections were performed in triplicates. dsODN was transfected as a negative control. Cells were harvested ~96 hr post-transfection and genomic DNA was purified. Efficient indel formation at the on-target site and integration of the dsODN tag were confirmed through deep amplicon sequencing. Genomic DNA was subsequently sheared with Covaris E220 to on average 500 bp according to the manufacturer's protocol. Sample libraries were assembled as previously described (*Tsai et al., 2015*) and sequenced using the Illumina MiSeq platform. Data were analysed using open-source guideseq software (version 1.1) (*Tsai et al., 2016*). Consolidated reads were mapped to the mouse mm39 reference genome. Upon identification of the genomic regions integrating the dsODNs in the aligned data, off-target sites with a maximum of six mismatches to the on-target site, which were absent in the background controls, were retained. A summary of the GUIDE-seq results can be found in **Supplementary file 8**.

## AAV production

Pseudo-typed vectors (AAV2 serotype PHP.eB) were produced by the Viral Vector Facility of the Neuroscience Center Zurich. Briefly, AAV vectors were ultracentrifuged and diafiltered. Physical titers (vector genomes per milliliter, vg/mL) were determined using a Qubit 3.0 fluorometer (Thermo Fisher Scientific) as previously published (*Düring et al., 2020*). The identity of the packaged genomes of each AAV vector was confirmed by Sanger sequencing.

## Stereotactic injections in mice

Unless stated otherwise, adult female C57BL/6 J mice at P50-P60 were used to introduce a unilateral lesion in the medial forebrain bundle. Buprenorphine [0.1 mg/kg bodyweight], was administered to mice subcutaneously 30 min prior to surgery. Animals were anesthetized using isoflurane (5% isoflurane with 1000 mL/min in 100% $O_2$) and placed into a stereotaxic mouse frame on a warming surface to maintain body temperature. Anesthesia was maintained at 1.5–2.5% isoflurane with 400 mL/min in 100% $O_2$ during surgeries. Mice were pre-treated with desipramine [25 mg/kg bodyweight] and pargyline [5 mg/kg bodyweight] 30 min before the injection of 6-hydroxydopamine (6-OHDA) was performed. 6-OHDA was dissolved in 0.02% ascorbate/saline solution at a concentration of 15 mg/mL and used within a maximum of 3 hr. 3.6 µg of 6-OHDA were injected into the medial forebrain bundle (mfb) at the following coordinates (relative to bregma): –1.2 mm anteroposterior (AP); 1.3 mm mediolateral (ML); –5 mm dorsoventral (DV). Sham-injected mice were injected with 0.02% ascorbate/saline solution. Injections were performed using a 5 µL Hamilton syringe with a 33 G needle at a speed of 0.05 µL/min. The needle was slowly removed 3 min after the injection and the wound was sutured using Vicryl 5–0 suture (Ethicon). Animals with unilateral lesions received extensive post-operative care for 2 wk. After the lesion, animals received daily glucose injections, and kitten milk (Royal Canin) for 1 wk to support recovery.

4–5 wk after the introduction of the 6-OHDA lesion, AAVs were injected into the substantia nigra, striatum, or visual cortex at the following coordinates (relative to bregma): –3.0 mm anteroposterior (A/P), 1.2 mm mediolateral (M/L), –4.5 mm dorsoventral (D/V) for the substantia nigra pars compacta; 0.38 mm A/P, 1.8 mm M/L, –4:0.4:–2.4 mm D/V for the striatum; and –4.5 mm A/P, 2.7 mm ML, 0.35 mm D/V for the visual cortex. Injections were performed using the same size needle, syringe, and speed as before. The needle was slowly removed 3 min after the injection and the wound was sutured using Vicryl 5–0 suture (Ethicon).

## Behavioral assays

Behavior experiments were performed at 4 wk after the 6-OHDA lesion and 12 wk after delivery of the treatment. Scientists performing and analyzing behavioral data were blinded during the study. To

analyze spontaneous rotations during the dark phase of the light cycle, mice were individually placed into a glass cylinder (10 cm diameter, 14 cm height), and after 1 min of habituation mouse behavior was recorded from the bottom using a digital camera. For assessment of spontaneous rotations after treatment, animals were first habituated to the experimental environment on three separate days. Full body ipsi- and contralateral turns (360°) were counted for 10 min. A frame-by-frame video player (VLC media player) was used for scoring. Data are expressed as a percentage of ipsilateral rotations from total rotations.

To assess forelimb akinesia during the light phase of the light cycle, we quantified left and right forelimb usage in the stepping test (*Blume et al., 2009*; *Olsson et al., 1995*). First, the animal was allowed to settle at one edge of the table (~2 s) with all limbs on the table. Next, the experimenter lifted the hind legs of the mouse by pulling up the tail, leaving only the forepaws touching the table. Animals were pulled backward by the tail at a steady pace of approximately 1 m in 3–4 s for a total distance of 1 m. Two trials of three consecutive repetitions were performed per animal with at least a 10 mi break between the two trials. Behavior was recorded from the side using a digital camera and the number of adjusting steps from both forepaws was counted. Data are represented as a percentage of ipsilateral steps from total steps.

For assessing drug-induced rotations, D-amphetamine (5 mg/kg bodyweight; Sigma-Aldrich) or apomorphine (0.5 mg/kg bodyweight; Sigma-Aldrich) was administered to mice via intraperitoneal injections. Following the injection, mice were placed in a recovery cage for 10 min. Afterward, mice were placed in a cylinder (10 cm diameter, 15 cm height) and habituated for 1 min. Rotations induced by D-amphetamine or apomorphine were recorded from the bottom for 10 min using a digital camera and only fully-body turns (360°) were counted as previously described. Data are expressed as a percentage of ipsilateral or contralateral rotations from total rotations.

## Trans-cardiac perfusion, brain isolation, and dissection of brain regions

Sodium pentobarbital (Kantonsapotheke Zürich) was injected via intraperitoneal injection at a dose of 100 mg/kg. Complete anesthesia was confirmed by the absence of a toe pinch reflex. Mice were placed on a perfusion stage inside a collection pan and the peritoneal cavity was exposed. The diaphragm was cut through laterally and the rib cage was cut parallel to the lungs, creating a chest 'flap.' The flap was clamped in place using a hemostat (Fine Science Tools) and a 25 G needle (Sterican), attached to silicon tubing and a peristaltic pump, was inserted into the left ventricle. The right atrium was cut for drainage. Animals were first perfused with ice-cold PBS (Thermo Fisher Scientific) at a rate of 10 mL/min, followed by perfusion with ice-cold fixative at the same rate (4% paraformaldehyde, PFA, Sigma-Aldrich). Once the perfusion was complete, mice were decapitated and the skull was removed with scissors and tweezers without inflicting damage to the underlying tissue. The brain was removed using a spatula.

For histology, PFA-perfused brains were post-fixated in 4% PFA for 4 hr, followed by overnight incubation in 30% sucrose. For neurotransmitter quantifications, brains were isolated, rinsed in PBS, and cut into 1 mm slices using an acrylic mouse brain matrix (AgnThos) and razor blades. The striatum was isolated under a stereomicroscope using the mouse brain atlas (*Paxinos and Franklin, 2001*). For amplicon sequencing, regions of interest (striatum or SNc) were manually dissected from 40 µm-thick coronal sections under a stereomicroscope using the mouse brain atlas (*Paxinos and Franklin, 2001*).

## Immunohistochemistry

Fresh or snap-frozen PFA-fixed brain tissues of C57BL/6 J mice were cut into 40µm-thick sections using a microtome. Sections were blocked in PBS supplemented with 5% normal donkey serum (cat. no. ab7475, abcam) and 0.3% Triton X-100 (Sigma-Aldrich) for 1 hr. Brain sections were incubated with primary antibodies overnight at 4 °C (rabbit-NEUN, 1:1'000, abcam 177487; mouse-TH; 1:1'000, Immunostar 22941; chicken-GFAP, 1:1'500, abcam ab95231; rat-BrdU, 1:400, Oxford Biotech OBT0030). Donkey anti-rabbit-488 (1:1'000), donkey anti-mouse-594 (1:500), donkey anti-chicken-647 (1:500), and donkey anti-rat-647 (1:500; all from Jackson ImmunoResearch) were used as secondary antibodies and sections were counterstained with 4',6-diamidino-2-phenylindole (DAPI, Sigma-Aldrich). Mounting was performed using Prolong Gold Antifade Mountant (Thermo Fisher Scientific). Images were taken with a Zeiss LSM 900 or a Zeiss AxioScan.Z1 slide scanner and analyzed with Fiji (*Schindelin et al., 2012*) or cell profiler (*Stirling et al., 2021*). The numerical density of cells was estimated using optical

dissection (NvVref method). Density of striatal fibres in the lesioned hemisphere was quantified as relative fluorescence intensity (FI) compared to the intact hemisphere. Additionally, the FI of the TH staining detected in the corpus callosum of each hemisphere was used for background correction of the FI detected in the striatum of the same hemisphere. Antibodies are listed in *Supplementary file 4*.

## Iterative immunofluorescence imaging (4i) and image analysis

Frozen PFA-fixed brain tissues of C57BL/6 J mice were cut into 40μm-thick sections using a microtome. Before mounting the sections, glass-bottomed 24-well plates (Cellvis P24-1.5H-N) were coated with poly-D-lysine (0.1 mg/mL, Sigma-Aldrich) for 5 min at RT on a shaker. Afterward, wells were rinsed three times with deionized water and left to dry overnight. Tissue sections were washed three times in PBS and transferred into the coated wells containing 500 μL of PBS, which was carefully aspirated with a glass pipette to allow the sections to adhere flat to the bottom. Sections were left to dry until there was no visible liquid remaining around the edges of the sections. Next, tissue sections were rinsed with PBS (3×5 min), followed by 1 hr incubation in blocking solution (PBS supplemented with 3% donkey serum, 0.5% Triton X-100, and 0.025% PFA) at RT. Sections were then incubated in primary antibodies (list in *Supplementary file 4*), diluted in blocking solution, for three nights at 4 °C. Next, sections were washed in PBS (3×5 min), rinsed in blocking solution for 5 min, followed by incubation with secondary antibodies and DAPI (1:1000; stock 1 mg/mL) for 2 hr at RT. All the following steps were performed under low light conditions to reduce possible fluorophore crosslinking. Last, sections were washed in PBS (3×5 min), and imaging buffer (PBS supplemented with N-Acetyl-cysteine at 0.7 M final concentration, pH 7.4) was added at least 5 min prior to imaging to guarantee penetrance of tissue sections. Once the imaging cycle had finished, sections were rinsed three times with $dH_2O$ and incubated in equal amounts of $dH_2O$ and elution buffer (3×5 min; 0.5 M L-glycine, 3 M urea, 3 M guanidine hydrochloride, 0.07 M TCEP-HCl; pH 2.5). Successful elution of each antibody was visually confirmed using a fluorescence microscope. After elution, tissue sections were washed three times in PBS (5 min) and then another 4i round was started. A total of four imaging cycles were performed.

All 4i z-stacks (image intervals of 0.5 μm) were collected on an ImageXpress Confocal HT confocal laser-scanning microscope with a 20 x water objective (NA 0.95) using bi-directional scanning. Samples of the same imaging cycle were labeled with the same antibodies and images were collected with identical microscopy settings for laser power, gain, digital offset, pinhole diameter, and z-step. Images from tile scans were exported using MetaXpress and analyzed using Fiji (*Schindelin et al., 2012*). DAPI intensity patterns were used to align image tiles from different staining cycles.

## *In vivo* BrdU proliferation assay

Five days after delivery of the treatment, bromodeoxyuridine was administered to mice at a concentration of 0.8 mg/mL via drinking water. Frozen PFA-fixed brain tissues of BrdU-treated mice were cut into 40 μm-thick sections using a microtome. Sections were washed 2×15 min and 1×5 min in PBS, followed by a 10 min incubation in 1 M HCl on ice, and a 25 min incubation in 2 M HCl at 37 °C. Next, tissues were rinsed in 0.1 M borate buffer (Sigma-Aldrich) for 10 min on a shaker at RT. After the tissues were rinsed in PBS for 6×10 min, brain tissues were stained as described in the section 'Immunohistochemistry'.

## Neurotransmitter purification and UHPLC-ECD quantifications

Snap-frozen fresh striata of lesioned or unlesioned hemispheres were used for the purification of neurotransmitters. All materials were kept cold on dry ice during the whole purification procedure and samples were kept under low light conditions on ice. Tissue samples were powderized with two pulses (at maximum intensity) using a CryoPrepTM system (Covaris). Equal amounts of powder were transferred to a pre-cooled 2 mL tube. For homogenization of the tissue powder, a metal ball (Qiagen) and 1 mL homogenization buffer (100 mM Tris-HCl, 2 mM EDTA, pH 7.6 supplemented with protease inhibitor tablet) were added to each tube and samples were homogenized for 2×90 s at 20 Hz using a TissueLyser II (Qiagen). Next, samples were centrifuged for 20 min at maximum speed and 4 °C. Lysates were next transferred to fresh pre-cooled tubes and 1 M HCl was added to a final concentration of 10% (v/v). Subsequently, lysates were filtered using an Amicon Ultra 0.5 (Sigma-Aldrich) and a table top centrifuge (30 min at 4 °C and maximum speed). 20 μL of the filtered sample was used for quantification of total protein amounts. Protein concentrations were determined using the

ABBOT Alinity C System (Abbot, Abbotpark, Illinois, USA, kit-no. 7P5920). 20 μL of the filtered sample was used in parallel for quantification of neurotransmitter levels by UHPLC-ECD. Brain monoamine neurotransmitter metabolites were analyzed in the filtered lysate using a modified Thermo Fisher Ulti-Mate 3000 High Sensitivity HPLC Electrochemical System (Thermo Fisher Scientific, Waltham, Massachusetts, USA). Injection volume of each sample was 20 μL and separation of the compounds was achieved using a YMC-Hydrosphere UHPLC column (C18 12 nm, S-2.0 μm, 150×30 mm, YMC Inc, Wilmington, NC, USA). As a mobile phase, a 56.7 mM sodium phosphate buffer, containing 5 mM octanesulphonic acid, 50 μM EDTA, 0.28% phosphoric acid (85%), and 23% methanol (pH 2.9–3.1, adjusted with concentrated 10 M NaOH), was used with an isocratic flow rate of 410 μL/min. The column was maintained at 27 °C by a surrounding TCC-3000SD column-thermostat. The analytical cell (Coulometric Cell Model 6011RS, Thermo Fisher Scientific) within the electrochemical detector ECD-3000RS (Thermo Fisher Scientific) was adjusted to a 10 mV potential and 100μA gain range for the upstream electrode, plus 400 mV potential, plus 500nA gain range for the downstream electrode with a response time of 1 s. Data was analyzed using the Chromeleon Chromatography Data System (CDS) Software 7.1.9 (Thermo Fisher) and corrected for the protein concentrations of the respective homogenates. All measurements were performed at the clinical chemistry unit of the Kinderspital Zürich.

## Statistical analysis

All statistical analyses were performed using GraphPad Prism 10.2.0 for macOS. If not stated otherwise, data are represented as biological replicates and are depicted as means ± standard deviation (s.d.). The sample size was approximated based on so-called Fermi methods and experience from previous base editing studies during experimental design. Sample sizes and the statistical analyses performed are described in the respective figure legends. Data were tested for normality using the Shapiro-Wilk test if not stated otherwise. For all analyses, a p-value of $p < 0.05$ was considered statistically significant. A detailed statistical report is provided as *Supplementary file 7*.

## Acknowledgements

We thank the Functional Genomics Center Zurich (FGCZ) for technical support and access to instruments at the University of Zurich. We thank Cornelia Schwerdel for technical support during *in vitro* and staining experiments. Annina Denoth Lipuner and Sebastian Jessberger are acknowledged for help with planning and setting up 4i experiments and sharing antibodies. Members of the Schwank, Patriarchi, and Häberle labs are acknowledged for discussions and comments on the manuscript. This work was supported by the Swiss National Science Foundation (SNSF) grant no. 310030_185293 (to GS) and 310030_196455 (to TP), Novartis Foundation for Medical-Biological Research no. FN20-0000000203 (to DB), SNSF Spark fellowship no. 196287 (to DB), the URPP Itinerare (to GS and to DB), the Helmut Horten Foundation (to GS), and the European Research Council (ERC) under the European Union's Horizon 2020 research and innovation program (grant agreement: 891959; to TP).

## Additional information

### Competing interests

Gerald Schwank: G.S. is an advisor to Prime Medicine. The other authors declare that no competing interests exist.

### Funding

| Funder | Grant reference number | Author |
| --- | --- | --- |
| Swiss National Science Foundation | 310030_185293 | Gerald Schwank |
| Helmut Horten Stiftung | | Gerald Schwank |
| Swiss National Science Foundation | 310030_196455 | Tommaso Patriarchi |

| Funder | Grant reference number | Author |
|---|---|---|
| Novartis Foundation for Medical-Biological Research | FN20-0000000203 | Desiree Böck |
| Swiss National Science Foundation | 196287 | Desiree Böck |
| European Research Council | 891959 | Tommaso Patriarchi |
| URPP Itinerare | | Desiree Böck Gerald Schwank |

The funders had no role in study design, data collection and interpretation, or the decision to submit the work for publication.

### Author contributions

Desiree Böck, Conceptualization, Formal analysis, Supervision, Funding acquisition, Validation, Investigation, Visualization, Methodology, Writing – original draft, Project administration, Writing – review and editing; Maria Wilhelm, Conceptualization, Formal analysis, Supervision, Validation, Investigation, Methodology, Project administration, Writing – review and editing; Jonas Mumenthaler, Daniel Fabio Carpanese, Formal analysis, Validation, Investigation, Writing – review and editing; Peter I Kulcsár, Formal analysis, Validation, Investigation, Visualization, Writing – review and editing; Simon d'Aquin, Validation, Investigation, Writing – review and editing; Alessio Cremonesi, Anahita Rassi, Johannes Häberle, Formal analysis, Writing – review and editing; Tommaso Patriarchi, Gerald Schwank, Conceptualization, Resources, Data curation, Formal analysis, Funding acquisition, Writing – review and editing

### Author ORCIDs

Desiree Böck ⓘ https://orcid.org/0000-0001-8142-0629
Tommaso Patriarchi ⓘ https://orcid.org/0000-0001-9351-3734
Gerald Schwank ⓘ https://orcid.org/0000-0003-0767-2953

### Ethics

Animal experiments were performed in accordance with protocols approved by the Kantonales Veterinäramt Zürich and in compliance with all relevant ethical regulations (license number ZH189/2020). C57BL/6J mice were housed in a pathogen-free animal facility at the Institute of Pharmacology and Toxicology of the University of Zurich. Mice were kept in a temperature- and humidity-controlled room on a 12-hr light-dark cycle. Mice were fed a standard laboratory chow (Kliba Nafag no. 3437 with 18.5% crude protein) with ad libitum access to food and water. Exclusion criteria were pre-defined during study design to meet ethical regulations. No animal was excluded from the study.

Reviewer #2 (Public review): https://doi.org/10.7554/eLife.97180.3.sa1
Reviewer #3 (Public review): https://doi.org/10.7554/eLife.97180.3.sa2
Author response https://doi.org/10.7554/eLife.97180.3.sa3

## Additional files

### Supplementary files

- Supplementary file 1. List of oligos used for cloning of sgRNA plasmids.
- Supplementary file 2. List of oligos used for cloning of adeno-associated virus (AAV) plasmids.
- Supplementary file 3. List of oligos used for RT-qPCR.
- Supplementary file 4. List of antibodies used in this study.
- Supplementary file 5. List of oligos used for deep sequencing.
- Supplementary file 6. Reference nucleotide sequences of amplicons for deep sequencing.
- Supplementary file 7. Statistic report of the presented data.
- Supplementary file 8. Summary of GUIDE-seq results in N2a cells.

• MDAR checklist

## Data availability

All data associated with this study are present in the paper. Illumina sequencing data is available under accession number GSE237570 at the Gene Expression Omnibus (GEO) data repository and BioProject number PRJNA1155634 at the NCBI Sequence Read Archive.

The following datasets were generated:

| Author(s) | Year | Dataset title | Dataset URL | Database and Identifier |
|---|---|---|---|---|
| Böck D, Wilhelm M, Patriarchi T, Schwank G | 2024 | Base editing of Ptbp1 in neurons alleviates symptoms in a mouse model of Parkinson's disease | https://www.ncbi.nlm.nih.gov/geo/query/acc.cgi?acc=GSE237570 | NCBI Gene Expression Omnibus, GSE237570 |
| Böck D, Wilhelm M, Patriarchi T, Schwank G | 2024 | Base editing of Ptbp1 in neurons alleviates symptoms in a mouse model of Parkinson's disease | https://www.ncbi.nlm.nih.gov/bioproject/PRJNA1155634/ | NCBI Sequence Read Archive, PRJNA1155634 |

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
