## [Editor Report · eLife Assessment]

This is an **important** study suggesting that neuron-specific loss of function of the RNA splicing factor Ptbp1 in striatal neurons induces dopaminergic markers and alleviates motor defects in a 6-hydroxydopamine (6-OHDA) mouse model of Parkinson's Disease. The evidence supporting the rescue of motor deficits following Ptbp1 manipulation is **solid**, and, while additional characterization of dopaminergic neuronal identity may be required in future studies, these results have clear implications for Parkinson's disease therapeutics. The study also addresses recent controversial literature on cell reprogramming in Parkinson's Disease and will be of interest to researchers with a focus on the application of gene therapy to rescue neurodegeneration.

---

## [Referee Report · Reviewer #2 (Public review)]

Summary:

The manuscript by Bock and colleagues describes generation of an AAV-delivered adenine base editing strategy to knockdown PTBP1 and the behavioral and neurorestorative effects of specifically knocking down striatal or nigral PTBP1 in astrocytes or neurons in a mouse model of Parkinson disease. The authors found that knocking down PTBP1 in neurons, but not astrocytes, and in striatum, but not nigra, results in the phenotypic reorganization of neurons to TH+ cells sufficient to rescue motor phenotypes, though insufficient to normalize responses to dopaminomimetic drugs.

Strengths:

The manuscript is well-written and adds to the growing literature challenging previous findings by Qian et al., 2020 and Zhou et al., 2020 indicating that astrocytic downregulation of PTBP1 can induce conversion to dopaminergic neurons in the midbrain and improve parkinsonian symptoms. The base editing approach is interesting and potentially more therapeutically relevant than previous approaches.

Weaknesses:

The animal model utilized, the 6-OHDA model, though useful to examine dopaminergic cell loss, exhibits accelerated neurodegeneration and none of the typical pathological hallmarks (synucleinopathy, Lewy bodies, etc.) compared to the typical etiology of Parkinson disease, limiting its translational interpretation. The identity of the converted neurons is unclear. Though the immunohistochemical methodology indicates they may be MSNs and/or interneurons, a more comprehensive identity is still lacking. There remains no real evidence that these cells actually release dopamine. Since striatal dopamine was assessed by whole-tissue analysis, which is not necessarily reflective of synaptic dopamine availability, it is difficult to assess whether the ~10% increase in TH+ cells in the striatum was sufficient to improve dopamine function. However, the improvement in motor activity suggests that it was.

---

## [Referee Report · Reviewer #3 (Public review)]

This study explores the use of an adenine base editing strategy to knock down PTBP1 in astrocytes and neurons of a Parkinson's disease mouse model, as a potential AAV-BE therapy. The results indicate that editing Ptbp1 in neurons, but not astrocytes, leads to the formation of tyrosine hydroxylase (TH)+ cells, rescuing some motor symptoms.

Several aspects of the manuscript stand out positively. Firstly, the clarity of the presentation. The authors communicate their ideas and findings in a clear and understandable manner, making it easier for readers to follow.

The Materials and Methods section is well-elaborated, providing sufficient detail for reproducibility.

The logical flow of the manuscript makes sense, with each section building upon the previous one coherently.

The ABE strategy employed by the authors appears sound, and the manuscript presents a coherent and well-supported argument.

Positively, some of the data in this study effectively counteracts previous work in line with more recent publications, demonstrating the authors' ability to contribute to the ongoing conversation in the field.

Comments on revisions:

The authors have adequately addressed all the previous questions and suggestions by providing new data and/or adding necessary clarifications and deeper discussions. The newly presented data convincingly fills the gaps identified during the initial review process. The additional discussions and clarifications enhance both the clarity and transparency of the manuscript.

---

## [Author Response]

The following is the authors’ response to the original reviews.

**Public Reviews:**

**Reviewer #1 (Public Review):**
Summary:Recent years have seen spectacular and controversial claims that loss of function of the RNA splicing factor Ptbp1 can efficiently reprogram astrocytes into functional neurons that can rescue motor defects seen in 6-hydroxydopamine (6-OHDA)-induced mouse models of Parkinson's disease (PD). This latest study is one of a series that fails to reproduce these observations, but remarkably also reports that neuronal-specific loss of function of Ptbp1 both induces expression of dopaminergic neuronal markers in striatal neurons and rescues motor defects seen in 6-OHDA-treated mice. The claims, if replicated, are remarkable and identify a straightforward and potentially translationally relevant mechanism for treating motor defects seen in PD models. However, while the reported behavioral effects are strong and were collected without sample exclusion, other claims made here are less convincing. In particular, no evidence that Ptbp1 loss of function actually occurs in striatal neurons is provided, and the immunostaining data used to claim that dopaminergic markers are induced in striatal neurons is not convincing. Furthermore, no characterization of the molecular identity of Ptbp1-deficient striatal neurons is provided using single-cell RNA-Seq or spatial transcriptomics, making it difficult to conclude that these cells are indeed adopting a dopaminergic phenotype.Overall, while the claims of behavioral rescue of 6-OHDA-treated mice appear compelling, it is essential that these be independently replicated as soon as possible before further studies on this topic are carried out. Insights into the molecular mechanisms by which neuronalspecific loss of function of Ptbp1 induces behavioral rescue are lacking, however. Moreover, the claims of induction of neuronal identity in striatal neurons by Ptbp1 require considerable additional work to be convincing.

We thank the reviewer for the detailed analysis of our study. Please find our answers to the points raised by the reviewer below in blue.

Strengths of the study:(1) The effect size of the behavioral rescue in the stepping and cylinder tests is strong and significant, essentially restoring 6-OHDA-lesioned mice to control levels.(2) Since the neurotoxic effects of 6-OHDA treatment are highly variable, the fact that all behavioral data was collected blinded and that no samples were excluded from analysis increases confidence in the accuracy of the results reported here.

We appreciate the reviewer’s feedback and acknowledgement of the strengths of our study. We undertook several optimization steps in the surgery, post-operative care, and handling of the animals for behavior experiments to ensure high reproducibility of our experiments.

Weaknesses of the study:(1) Neurons express relatively little Ptbp1. Indeed, cellular expression levels as measured by scRNA-Seq are substantially below those of astrocytes and other non-neuronal cell types, and Ptbp1 immunoreactivity has not been observed in either striatal or midbrain neurons (e.g. Hoang, et al. Nature 2023). This raises the question of whether any recovery of Th expression is indeed mediated by the loss of function of Ptbp1 rather than by off-target effects. AAVmediated rescue of Ptbp1 expression could help clarify this.

In the original manuscript, we delivered control vectors that only express the ABE to 6-OHDAlesioned mice (labeled as AAV-ctrl) and did not detect TH positive cells in the midbrain or striatum of control mice or rescue of spontaneous motor skills. We can therefore exclude that the delivery procedure, AAV-PHP.eB capsid, or ABE expression caused adverse effects leading to induction of TH expression and functional rescue of spontaneous motor behaviors in PD mice. To further exclude that these effects were caused by off-target editing, we experimentally determined off-target binding sites of our sgRNA (sgRNA-ex3) using GUIDEseq and subsequently analyzed these sites in treated animals by NGS (Figure 3 – supplement 3). While two off-target sites were identified, it is unlikely that base editing at these sites caused the observed phenotypes. One off-target site was identified in the myopalladin (Mypn) gene, which encodes for a muscle-specific protein that plays a role in regulating the structure and growth of skeletal and cardiac muscle (Filomena et al., 2021, 2020). The other site is not located in a coding region, but in an intron of the ankyrin-1 (Ank1) gene, encoding for an adaptor protein linking membrane proteins to the underlying cytoskeleton (Cunha and Mohler, 2009). Even though this gene is also expressed in neurons, base editing within this intronic region did not lead to changes in transcript levels (Figure 3 – supplement 3). Thus, the induction of TH expression upon adenine base editing with sgRNA-ex3 is likely a direct consequence of PTBP1 downregulation.

Further supporting this conclusion, in the revised manuscript we additionally show PTBP1 downregulation at the RNA and protein level in the SNc and striatum after base editor treatment (Figure 2 – figure supplement 5; figure 3 – supplement 2).

(2) It is not clear why dopaminergic neurons, which are not normally found in the striatum, are observed following Ptbp1 knockout. This is very similar to the now-debunked claims made in Zhou, et al. Cell 2020, but here performed using the hSyn rather than GFAP mini promoter to control AAV expression. While this is the most dramatic and potentially translationally relevant claim of the study, this claim is extremely surprising and lacks any clear mechanistic explanation for why it might happen in the first place.

We agree with the reviewer that our study does not provide mechanistic insights into how Ptbp1 downregulation in neurons leads to the induction of dopaminergic markers in the striatum. As we believe that this is not within the scope of a revision, we discuss potential follow-up experiments in the discussion section of the revised manuscript.

This observation is even more surprising in light of reports that antisense oligonucleotidemediated knockdown of Ptbp1, which should have affected both neuronal and glial Ptbp1 expression, failed to induce expression of dopaminergic neuronal markers in the striatum (Chen, et al. eLife 2022). Selective loss of function of Ptbp1 in striatal and midbrain astrocytes likewise results in only modest changes in gene expression.

Using 6-OHDA lesioned *Aldh1l1-CreERT2;Rpl22lsl-HA* mice, the Chen et al. study (eLife 2022) assessed potential astrocyte to neuron conversion by quantifying the presence of HA-labeled neurons after ASO-mediated knockdown of Ptbp1. Even though they did not detect HApositive neurons in the SNc, suggesting absence of astrocyte to neuron conversion, the images in Figure 4D reveal TH positive cells in the lesioned hemisphere, similar to our observations in Figure 2B-D. While it cannot be excluded that these TH positive cells are remnants from an incomplete 6-OHDA lesion, they could also be endogenous neurons with induced expression of dopaminergic markers after ASO-mediated knockdown of Ptbp1. Furthermore, Chen et al. performed the apomorphine test to assess changes in motor skills, which did not reveal an improvement in our study either.

It is critically important that this claim be independently replicated, and that additional data be provided to conclusively show that striatal neurons are indeed expressing dopaminergic markers.

Our behavior and immunofluorescence experiments involving mice injected into the striatum were performed with two independently generated cohorts of 6-OHDA mice. In detail, the 6OHDA mice were generated by two independent surgeons from different labs (>6 months between experiments of these cohorts), leading to comparable behavioral outcomes before and after treatment. Subsequent behavior and immunofluorescence experiments with each cohort were performed and analyzed by two independent and blinded researchers, showing comparable results.

(3) More generally, since multiple spectacular and irreproducible claims of single-step glial-toneuron reprogramming have appeared in high-profile journals in recent years, a consensus has emerged that it is essential to comprehensively characterize the identity of "transformed" cells using either single-cell RNA-Seq or spatial transcriptomics (e.g. Qian, et al. FEBS J 2021; Wang and Zhang, Dev Neurobiol 2022). These concerns apply equally to claims of neuronal subtype conversion such as those advanced here, and it is essential to provide these same datasets.

In the revised version, we have analyzed the expression of additional neuronal markers in TH positive cells of the striatum using 4i imaging. Briefly, our results showed that the vast majority of TH-expressing cells also expressed the markers DAT and NEUN, further corroborating the neuronal and dopaminergic identity of these cells. Additional analysis revealed that this TH/DAT/NEUN expressing cell population expressed markers of GABAergic neurons, either of medium spiny neurons (~50%) and various types of interneurons (~50%). While our 4i analysis has allowed us to broadly classify these TH-expressing populations, we agree that detailed transcriptional analysis at the single cell level is required to understand the molecular mechanisms underlying the generation of TH positive cells. These analyses are, however, not within the scope of a revision and would require a thorough dedicated study. We have added these results and discussion points to the revised manuscript.

(4) Low-power images are generally lacking for immunohistochemical data shown in Figures 3 and 4, which makes interpretation difficult. DAPI images in Figure 3C do not appear nuclear. Immunostaining for Th, DAT, and Dcx in Figure 4 shows a high background and is difficult to interpret.

We thank the reviewer for closely evaluating these images and suggestions for improvement. In the revised manuscript, we provide low power images and higher magnification insets as requested to allow for easier interpretation.

(5) Insights into the mechanism by which neuronal-specific loss of Ptbp1 function induces either functional recovery, or dopaminergic markers in striatal neurons, is lacking.

In the revised manuscript, we provide a more detailed discussion of mechanisms that could potentially be involved in the functional recovery or expression of dopaminergic markers. However, deciphering the exact molecular mechanisms underlying these observations requires thorough transcriptional analysis at the single cell level, which is out of scope of this revision.

**Reviewer #2 (Public Review)**:Summary:The manuscript by Bock and colleagues describes the generation of an AAV-delivered adenine base editing strategy to knockdown PTBP1 and the behavioral and neurorestorative effects of specifically knocking down striatal or nigral PTBP1 in astrocytes or neurons in a mouse model of Parkinson's disease. The authors found that knocking down PTBP1 in neurons, but not astrocytes, and in striatum, but not nigra, results in the phenotypic reorganization of neurons to TH+ cells sufficient to rescue motor phenotypes, though insufficient to normalize responses to dopaminomimetic drugs.Strengths:The manuscript is generally well-written and adds to the growing literature challenging previous findings by Qian et al., 2020 and Zhou et al., 2020 indicating that astrocytic downregulation of PTBP1 can induce conversion to dopaminergic neurons in the midbrain and improve parkinsonian symptoms. The base editing approach is interesting and potentially more therapeutically relevant than previous approaches.Weaknesses:The manuscript has several weaknesses in approach and interpretation. In terms of approach, the animal model utilized, the 6-OHDA model, though useful to examine dopaminergic cell loss, exhibits accelerated neurodegeneration and none of the typical pathological hallmarks (synucleinopathy, Lewy bodies, etc.) compared to the typical etiology of Parkinson's disease, limiting its translational interpretation.

We thank the reviewer for the detailed assessment of our study and pinpointing its current weaknesses. Please find our answers to all comments below in blue.

We agree with the reviewer that the 6-OHDA model lacks the typical pathological hallmarks of PD. Nevertheless, we chose this model for two reasons:

i) The 6-OHDA model was used by both Qian et al. (2020) and Zhou et al. (2020). To allow comparison of our results to these studies, it was crucial to use the same model. Notably, the 6-OHDA model was also used by Chen et al. (2022) and Hoang et al. (2023) for comparison to the two studies from 2020.

ii) The 6-OHDA model is straightforward to generate and displays robust motor impairments for evaluation of potential therapeutic effects of neuroregeneration treatment approaches. We therefore believe that the model is well-suited to analyze the cellular and behavioral effects (specifically motor skills) of PTBP1 downregulation.

In future studies, it would be critical to include models that also display typical pathological hallmarks of the disease to further evaluate the therapeutic effect of this base editing approach. These experiments are, however, not within the scope of this study, which was aimed to focus on the cellular and behavioral effects of PTBP1 downregulation.

In addition, there is no confirmation of a neuronal or astrocytic knockdown of PTBP1 in vivo; all base editing validation experiments were completed in cell lines.

In the revised manuscript, we assess *in vivo* base editing efficiencies at the *Ptbp1* target site in the SNc (AAV-hsyn, 15.6%) and striatum (AAV-hysn, 21.1%). Furthermore, we assessed *in vivo Ptbp1* downregulation at the RNA and protein level to complement our *in vitro* data (Figure 2 – figure supplement 5; figure 3 – supplement 2).

Finally, it is unclear why the base editing approach was used to induce loss-of-function rather than a cell-type specific knockout, if the goal is to assess the effects of PTBP1 loss in specific neurons.

We expressed base editors under cell-type specific promoter to induce a reliable loss-offunction mutation at the *Ptbp1* exon-intron junction in neurons or astrocytes. Performing these mutations with Cas9 nucleases instead would have had potential limitations and risks, including (i) indel mutations do not always lead to a frameshift and loss-of-function despite high indel formation at the targeted site, (ii) nucleases induce DNA double strand breaks, which can have serious side effects (e.g. chromosomal rearrangements or translocations), and (iii) ‘mosaicisms’ as edited cells contain different indel mutations, which may result in different effects and thus complicate analysis of the downstream effects. We discuss these points in the revised manuscript.

In terms of interpretation, the conclusion by the authors that PTBP1 knockdown has little likelihood to be therapeutically relevant seems overstated, particularly since they did observe a beneficial effect on motor behavior. We know that in PD, patients often display negligible symptoms until 50-70% of dopaminergic input to the striatum is lost, due to compensatory activity of remaining dopaminergic cells. Presumably, a small recovery of dopaminergic neurons would have an outsized effect on motor ability and may improve the efficacy of dopaminergic drugs, particularly levodopa, at lower doses, averting many problematic side effects. Since striatal dopamine was assessed by whole-tissue analysis, which is not necessarily reflective of synaptic dopamine availability, it is difficult to assess whether the ~10% increase in TH+ cells in the striatum was sufficient to improve dopamine function. However, the improvement in motor activity suggests that it was.

As pointed out by the reviewer, it is difficult to estimate the therapeutic effect and importance of a ~10% increase in TH+ cells for PD patient. Guided by the reviewer’s suggestion, we have included a more in-depth discussion of our results and its potential therapeutic value as well as outstanding questions for future studies in the revised manuscript.

**Reviewer #3 (Public Review):**
This study explores the use of an adenine base editing strategy to knock down PTBP1 in astrocytes and neurons of a Parkinson's disease mouse model, as a potential AAV-BE therapy. The results indicate that editing Ptbp1 in neurons, but not astrocytes, leads to the formation of tyrosine hydroxylase (TH)+ cells, rescuing some motor symptoms.Several aspects of the manuscript stand out positively. Firstly, the clarity of the presentation. The authors communicate their ideas and findings in a clear and understandable manner, making it easier for readers to follow.The Materials and methods section is well-elaborated, providing sufficient detail for reproducibility.The logical flow of the manuscript makes sense, with each section building upon the previous one coherently.The ABE strategy employed by the authors appears sound, and the manuscript presents a coherent and well-supported argument.Positively, some of the data in this study effectively counteracts previous work in line with more recent publications, demonstrating the authors' ability to contribute to the ongoing conversation in the field.

We thank the reviewer for appreciating the effort we have put into this study. Please find below a point-by-point reply to the weaknesses raised by the reviewer.

However, while the in vitro data yields promising results, it may have been overly optimistic to assume that the efficiencies observed in dividing cells will directly translate to in vivo conditions. This consideration is important given the added complexities of vector optimization, different cell types targeted in vitro versus in vivo, as well as unknown intrinsic limitations of the base editing technology.

We agree with the reviewer that *in vitro* base editing efficiencies might not directly translate to *in vivo* editing outcomes. We therefore assessed *in vivo* base editing efficiencies at the *Ptbp1* locus and PTBP1 downregulation in the striatum and midbrain. Our data revealed that *in vivo* base editing activity was lower than in our *in vitro* setting (*in vitro*: Figure 1; figure 1 – figure supplement 2; *in vivo*: figure 2 – figure supplement 5; figure 3 – supplement 2). However, we believe that these rates are slightly underestimated since we sequenced DNA isolated from the whole tissue (striatum or SNc) and not from purified astrocytes or neurons. Moreover, we could demonstrate that editing led to a reduction of *Ptbp1* transcript and PTBP1 protein level (Figure 2 – figure supplement 5; figure 3 – supplement 2).

In addition, certain aspects of the manuscript would benefit from a more in-depth and comprehensive discussion rather than being only briefly touched upon. Such a discussion would enhance the relevance of the obtained results and provide the foundation for improvement when using similar approaches.

Following the reviewer’s suggestion, we included a more in-depth discussion of our results in the revised manuscript.

**Recommendations for the authors:**

**Reviewing Editor (Recommendations for the Authors):**
A summary of key recommendations that might improve the eLife assessment in a subsequent submission are provided below, as a guide to help the authors focus on changes that might enhance the strength of evidence (e.g., from "incomplete" to "solid").(1) Provide further explanation of the mechanistic relationship between the downregulation of Ptbp1 and TH+ dopaminergic neuron reprogramming. Additional discussion of this topic should also be included.(2) Demonstrate proof of editing in the intended targeted cells in vitro and/or in vivo.(3) Show evidence of successful Base Editor delivery in vivo.(4) Perform a deeper characterization of TH+ cells in vivo and provide a more thorough discussion of the identity of the targeted cells. This may include an exploration of whether TH+ cells detected are TH+ interneurons and/or establish their identity based on transcriptomics or a similar approach.(5) Provide better-quality representative images supporting the quantitative data.(6) Please include full statistical reporting including exact p-values wherever possible alongside the summary statistics (test statistic and df) and 95% confidence intervals. These should be reported for all key questions and not only when the p-value is less than 0.05 in the main manuscript.

In the revised manuscript, we provided (1) suggestions of the mechanistic relationship between Ptbp1 knockdown, dopamine synthesis, and the functional rescue of spontaneous behaviors, (2) proof of *in vivo* base editing and successful base editor delivery, (3) deeper characterization of TH-expressing cells *in vivo* using 4i imaging, (4) better quality images, and (5) full statistical reporting.

Individual Reviewer recommendations for the authors are included below.
**Reviewer #1 (Recommendations For The Authors):**
Confirm loss of Ptbp1 function in infected striatal neurons. Single-cell RNA-Seq or spatial transcriptomic analysis must be performed to characterize the identity of the edited striatal neurons. The quality of the immunostaining in Figures 3 and 4 needs to be improved, and lowpower images provided. Were eLife a conventional journal, I would have insisted on all these being included prior to publication. Please also arrange for independent replication of the behavioral rescue and induction of dopaminergic marker gene expression in the striatum.

In the revised manuscript, we confirmed *Ptbp1* downregulation at the tissue level in the SNc and striatum by RT-qPCR and western blot and included low-power images for easier interpretation. Additionally, we assessed expression of additional neuronal markers on striatal sections using 4i imaging and found that TH/DAT/NEUN positive populations either expressed markers of medium spiny neurons or interneurons. We have included these results in the revised manuscript.

Our behavioral and imaging experiments involving mice injected into the striatum were in fact performed with two independently generated cohorts of 6-OHDA mice. In detail, the 6OHDA mice were generated by two independent surgeons from different labs (>6 months between experiments of these two cohorts), leading to comparable behavioral outcomes before and after treatment. The experiments with each cohort were performed and analyzed by two independent and blinded researchers, yielding comparable results.

**Reviewer #2 (Recommendations For The Authors):**
(1) In the introduction, lines 43-45: This statement is inaccurate. Current treatment strategies do not focus on slowing or halting disease progression. There is currently no accepted therapy that does this. Dopaminergic therapies and deep brain stimulation can compensate for circuitry dysfunction as a result of dopamine cell loss but do not slow the disease. The referenced paper used is older and does not refer to new treatments for PD and is a summary article for a special issue of the Disease Models and Mechanisms journal. Please ensure that all references used are appropriate for the statement they are attached to.

We thank the reviewer for pointing this out. We have rephrased this statement accordingly and provided an appropriate reference describing current treatment strategies.

(2) The number of TH+ cells in the intact nigra seems low compared to published data. Suggest a stereological approach may be better than the Abercrombie method.

Following the reviewer’s suggestion, we re-quantified the number of TH positive cells using a stereological approach (Nv:Vref method). We have included these results in the revised manuscript.

(3) Have the authors considered that the striatal TH+ cells could be TH+ striatal interneurons?

In the revised manuscript, we performed additional 4i imaging experiments to further analyze the identity of the TH positive cells in the striatum. Briefly, we found that TH/DAT/NEUN positive populations either expressed markers of GABAergic medium spiny neurons or interneurons. We have added these results to the revised manuscript (Figure 4).

(4) The Western blot shown in Figure 1 C for C8-D1A has some abnormalities and makes it difficult to judge the bands. Also, for 1B, the legends are difficult to see.

In the revised manuscript, we have repeated the respective western blot to make interpretation of the bands easier, and adapted the legends in Figure 1B for better visibility.

(5) Figure 2: Please show representative images for the GFAP-targeted editing.

Representative images of the GFAP-targeted groups can be found in Figure 2 – figure supplement 3.

(6) Figure 2, Supplement 3: Please include quantification.

The quantifications for these images can be found in Figure 2D and 2F.

(7) Figure 1, Supplement 2: The gene name in A is misspelled.

Thank you for point this out. In the revised manuscript, we added the correct gene name.

(8) Line 267-276: As previously indicated, the statement here is overstated based on the data provided. In addition, the citation provided to justify this claim (Kannari et al., 2000) is an odd choice as the dosage of L-DOPA utilized was not therapeutically relevant (50 mg/kg). A better indication of efficacy would be the return to basal, unaffected levels rather than the fold increase in dopamine levels. A better comparison would be Lindgren et al., 2010 who showed that L-DOPA-treated animals with a physiologically relevant dose (6 mg/kg) that did not induce dyskinesia, showed a return to basal, non-lesioned dopamine levels in the striatum after LDOPA by microdialysis. To really support this claim, the authors would need to use an approach that could measure synaptic dopamine availability, rather than whole-tissue dopamine levels, such as microdialysis, fiber photometry, or an equivalent.

Following the reviewer’s suggestions, we replaced this reference with Lindgren et al. (2010) and provide a more detailed interpretation of our results and remaining questions for future studies.

**Reviewer #3 (Recommendations For The Authors):**
Major and minor issues are discussed below by section.INTRODUCTION and AIM - Lines 36-73- The authors effectively contextualize the aim of their study by providing comprehensive background information on previous research regarding cell 'reprogramming' into dopaminergic neurons in the SNc. However, the introduction lacks contextualization of TH+ cells and PD. For readers who may not be well-versed in the Parkinson's field, understanding the importance of TH (Tyrosine Hydroxylase) may be challenging, since the term "TH+ cells" is mentioned only once by the end of the introduction (line 71), to then become a key element in the entire study.- Providing a brief explanation of the role of Tyrosine Hydroxylase in the synthesis of L-DOPA would facilitate the reader's comprehension of why the presence of TH+ cells following Base Editing treatment is relevant.- Further elaboration on the relationship between the downregulation of the general RNA binding protein, PTBP1, and the specific dopaminergic-related readout, TH, would improve coherence and strengthen the linkage between the introductory section and the results.

We thank the reviewer for the constructive suggestions. In the introduction of the revised manuscript, we describe the meaning and importance of TH in the context of dopamine synthesis and PD. Likewise, we briefly outlined the importance of the PTBP1/nPTBP regulatory loops during neuronal differentiation and maturation.

RESULTSResult Section 1 - Line 75-109- Thorough screening of sgRNAs targeting splice junctions across the Ptbp1 gene in HEPA cells, shows the achievement of high levels of editing (80-90%) with sgRNA-ex3 and sgRNAex7.- The data also indicates that editing translates into significant reductions in ptbp1 expression, along with an increase in the expression of genes repressed by PTBP1.- Despite obtaining lower percentages of editing events in N2a neuroblastoma cells and the C8-D1A astroglial cell line, the differential expression levels of ptbp1 and the readout genes remain significant. However, the gRNA screening assay is performed in immortalized, dividing cells.- Providing proof that Adenosine Base Editing of Ptbp1 is successful in non-dividing cells (such as SNc and/or striatal primary neurons) would strengthen the case for the potential therapy in the intended cell type.

Following the reviewer’s comment, we show *in vivo* base editing rates in the SNc and striatum of treated PD mice in the revised manuscript (Figure 2 – figure supplement 5; figure 3 – supplement 2).

- Moreover, assessing the expression levels of tyrosine hydroxylase by qPCR after Ptbp1 base editing in vitro could help contextualize the use of TH+ detection as an in vivo readout and may help explain why the total number of TH+ cells is low after ABE treatment in vivo - as shown in following sections.

In the revised manuscript, we now provide quantifications of *in vivo* base editing efficiencies in the SNc (~15%) and striatum (~20%). As expected from these lower *in vivo* base editing rates, downregulation of *Ptbp1* at the transcript and protein level was less pronounced compared to our *in vitro* experiments. It seems likely that higher base editing efficiency and more pronounced downregulation of *Ptbp1* could lead to a larger population of TH expressing cells. We have added these results and interpretations to the revised manuscript.

- Furthermore, although ABEs are less prone to generating bystander and other nucleotide changes compared to CBEs, it is still possible. Figures 1 (line 811) and 1-supplement 2 (line 842) only show a brief window of the Sanger sequencing trace. Updating these figures to display a wider view of the sequencing trace would enhance transparency. If unwanted edits are detected, while they may not significantly alter the relevance, impact, or structure of the paper, they may become an important aspect of the discussion.

Indeed, ABEs can induce bystander edits and we also detected such edits at the *Ptbp1* target site. However, since our base editing strategy was designed to yield a loss of *Ptbp1* function, bystander editing at the splice site was not a primary focus in our analysis. Nevertheless, we included CRISPResso output images showing the specific editing outcomes in a wider analysis window in the revised manuscript (Figure 3 – figure supplement 2).

Result Section 2 - Lines 110-159A split intein system is used in vivo with sgRNA-ex3, after updating the promoter to make it cell-specific: hSyn to restrict expression to neurons and GFAP to restrict expression to astrocytes.However, no other assay is performed to assess whether (a) the promoter change and/or (b) splitting Cas9 may affect the editing efficiency compared to their initial in vitro approach.

In the revised manuscript, we assessed the performance of the *in vivo* AAV vectors encoding the split intein ABE with sgRNA-ex3 *in vitro* in N2a and C8-D1A cells. Our results show that all vectors are functional and result in base editing at the target locus.

- Addressing whether this is the case may explain the low number of TH+ cells observed in vivo.- The authors could also consider staining for Cas9 to address whether the low number of TH+ cells could be attributed to a poor Cas9 delivery.

To confirm successful *in vivo* base editor delivery, we quantified *in vivo* base editing efficiencies in the SNc and striatum of PD mice. Our analysis revealed *in vivo* base editing efficiencies at both tissue sites, confirming that base editors were successfully delivered. Editing efficiencies were, however, substantially lower (Figure 2 – figure supplement 5; figure 3 – supplement 2). than in our *in vitro* cell line setting (Figure 1; figure 1 – figure supplement 2). Even though tissue editing rates likely underestimate the cell type-specific editing rates in astrocytes or neurons, higher base editing rates would have likely resulted in a higher number of TH positive cells. We have added these results and their implications to the revised manuscript.

- Moreover, despite the presence of TH, in Figure 2 E,F authors examine the striatal innervation from newly generated TH+ cells in the SNc by Fluorescence Intensity (FI) to conclude that the edited cells do not form projections towards the striatum. Considering the low levels of TH+ positive cells obtained, the accumulation of gross FI might not be the most accurate way to assess the presence or absence of cell projections.- Using another marker that stains the projections rather than the cell soma, and that is a marker of dopaminergic neurons, might be a better way to address this.

To address the reviewer’s comment, we analyzed the presence of potential dopaminergic fibers in the mfb, where projections are more concentrated (around the injection coordinates of 6-OHDA), using the dopaminergic marker DAT. In line with our previous observations in the striatum, we did not detect an increase in DAT fluorescence intensity upon treatment on the lesioned hemisphere (Figure 2 – figure supplement 4).

Result Section 3 - Line 160-182Minor issue- The same dual split intein system is used in the striatum. However, in Figure 3 - Figure Supplement 1 - line 958 and in Figure 3 - Figure Supplement 4 - line 1000authors show the injection of 2x the viral genomes indicated along the manuscript. In previous experiments the SNc 2x108vg/animal was used whereas this figure shows 4x108vg/animal injected in the striatum.- The authors should clarify if the vg injected in the striatum was different from what they previously indicated.

Compared to injection in the SNc, the volume of vector injected in the striatum was doubled since the region is significantly larger. We clarified that the injected vector genomes were different between striatum and SNc in the revised manuscript.

Result Section 4- Line 183-220In this section, the authors thoroughly examine the neuronal nature of TH+ cells through NeuN co-staining and iterative immunofluorescence imaging (4i). BrdU experiments are conducted to determine the origin of these cells, leading to the conclusion that TH+ cells derive from nondividing cells and express the neuronal marker DAT, characteristic of dopamine-producing neurons (DANs). Cell shape of the TH+ cells in the striatum and SNc is also evaluated measuring their Feret's diameter and their cell surface. Authors conclude there's heterogeneity in the TH+ cell population due to the presence of TH+/Neun- as well as differences in cell shape.However, their explanation of this heterogeneity is solely attributed to differences in the microenvironment and lacks further elaboration. Similarly, their observation that almost half the number of TH+ striatal cells after treatment express CTIP2 (Line 213 and Figure 4B), a marker for GABAergic medium spiny neurons, which they state as "interesting" (line 213) is not developed further. Delving deeper into these topics could strengthen the discussion.

In the revised manuscript, we provided a more in-depth discussion of the 4i imaging results and potential therapeutic implications. Additionally, we suggest follow-up experiments to analyze the identity, function, and molecular mechanisms underlying the expression of TH upon PTBP1 downregulation in future studies.

Result Section 5- Line 221-243Two drug-free and two drug-induced behavioral tests are conducted in control and treated animals to evaluate the restoration of motor functions following treatment. Consistent with their previous findings, only the treatment targeted to neurons resulted in the restoration of motor functions in drug-free behavioral tests. The rationale behind each test and its evaluation is clearly explained.DISCUSSION- In the discussion section, the authors effectively re-examine their results contextualizing their data with previous studies in the field. However, it would be helpful at this point in the manuscript to reconsider the use of the term 'cell reprogramming,' as this study does not involve actual cell reprogramming. The concept "reprograming" entails the process of transforming adult cells into a stem cell-like state, to then differentiate them into a different cell type. As proven in section 4 by a BrdU proliferation assay, the targeted cells are differentiated neurons. Considering BrdU is administered 5 days after ABE treatment, if true cell reprogramming was taking place, there should be evidence of BrdU incorporation. Cell reprogramming or reprograming is mentioned 4 times in the manuscript (line 34, line 54, line 265, line 277). Therefore, using another terminology would be more accurate.

Following the reviewer’s suggestion, we removed the term “cell reprograming” from the manuscript and rather describe it as induction of TH expression in endogenous neurons.

- As noted in the comments of section 4, a more thorough discussion about the various possibilities for heterogeneity would enhance the manuscript's contribution to the PD field.

In the revised manuscript, we provided a more in-depth discussion of the 4i imaging results and potential therapeutic implications.

- Despite observing low numbers of TH+ cells, no significant rescue of drug-induced behaviors, and low levels of released dopamine, the authors merely state that these results make the therapy non-viable, but there is no further exploration or discussion. Whether the limitations lie in the ABE strategy itself, such as its efficiency in targeting and editing of differentiated neurons; or if the issues lie on the injection and delivery, is never discussed. A deeper argumentation on the possible underlying reasons for these challenges would greatly enhance the manuscript and contribute to the advancement of ABE therapies in the brain.

We believe that the efficacy of our base editing approach could be significantly enhanced by optimizing the delivery. Currently, we are using a dual AAV approach to deliver intein-split ABEs. Since this approach relies on the delivery of higher AAV doses to achieve cotransduction of a cell by two different AAVs, the efficiency could be significantly enhanced by using smaller Cas9 orthologues that can be delivered as a single AAV. Furthermore, in this study we performed a single injection into the dorsal striatum to deliver ABE-expressing AAVs. Performing multiple injections into the rostral, medial, and caudal regions of the striatum might allow us to transduce more cells and induce TH expression in a larger population of striatal neurons. We have included these points in the revised manuscript.

- While drug-induced behaviors are not recovered, the data demonstrates a rescue of spontaneous behaviors. Further discussion on the potential differences in circuitry underlying these variations in behavioral rescue would also enrich the manuscript's discussion.

In the revised manuscript, we provide suggestions for potential mechanisms involved in the rescue of spontaneous behavior vs. absence of rescue of drug-induced behaviors.

FIGURES AND FIGURE SUPPLEMENTSGeneral minor issue - low magnification images in the following figures, make it difficult to visualize positive cells in tissue sections: Figure 2; Figure 2- supplement 1; Figure 2 - supplement 3, Figure 3- supplement 1. Adding a higher magnification imaging of positive cells in tissue sections of SNc and striatum might help with the visualization.

As suggested by the reviewer, we included higher magnification images in the corresponding figures to improve interpretation of our results.